# New Methodology Shows Short Atmospheric Lifetimes of Oxidized Sulfur and Nitrogen due to Dry Deposition

Katherine Hayden[1], Shao-Meng Li[1,2*], Paul Makar[1], John Liggio[1], Samar G. Moussa[1], Ayodeji Akingunola[1], Robert McLaren[3], Ralf M. Staebler[1], Andrea Darlington[1], Jason O'Brien[1], Junhua Zhang[1], Mengistu Wolde[4], Leiming Zhang[1]

[1]*Air Quality Research Division, Environment and Climate Change Canada, Toronto, Ontario, Canada, M3H 5T4*

[2]*College of Environmental Science and Engineering, Peking University, Beijing 100871 China*

[3]*Center for Atmospheric Chemistry, York University, 4700 Keele Street, Toronto, Ontario, Canada*

[4]*National Research Council Canada, Flight Research Laboratory, Ottawa, Canada K1A 0R6*

*corresponding authors shaomeng.li@pku.edu.cn; Katherine.hayden@canada.ca

For submission to *Atmospheric Chemistry and Physics*

**Abstract**

The atmospheric lifetimes of pollutants determine their impacts on human health, ecosystems and climate and yet, pollutant lifetimes due to dry deposition over large regions have not been determined from measurements.  Here, a new methodology based on aircraft observations is used to determine the lifetimes of oxidized sulfur and nitrogen due to dry deposition over $(3-6) \times 10^3$ $km^2$ of boreal forest in Canada.  Dry deposition fluxes decreased exponentially with distance from the Athabasca oil sands sources, located in northern Alberta, resulting in lifetimes of 2.2-26 hours.  Fluxes were 2-14 and 1-18 times higher than model estimates for oxidized sulfur and nitrogen, respectively, indicating dry deposition velocities which were 1.2-5.4 times higher than those computed for models.  A Monte-Carlo analysis with five commonly used inferential dry deposition algorithms indicates that such model underestimates of dry deposition velocity are typical.  These findings indicate that deposition to vegetation surfaces are likely under-estimated in regional and global chemical transport models regardless of the model algorithm used. The model-observation gaps may be reduced if surface pH, and quasi-laminar and aerodynamic resistances in algorithms are optimized as shown in the Monte-Carlo analysis.  Assessing the air quality and climate impacts of atmospheric pollutants on regional and global scales requires improved measurement-based understanding of atmospheric lifetimes of these pollutants.

## 1 Introduction

Deposition represents the terminating process for most air pollutants and the starting point for ecosystem impacts. Understanding deposition is critical in determining the atmospheric lifetimes and spatial scale of atmospheric transport of pollutants, which in turn, dictates their ecosystem (WHO, 2016; Solomon et al., 2007) and climate (Samset et al., 2014) impacts. In particular, atmospheric lifetimes ($\tau$) of oxidized sulfur and nitrogen compounds influence their concentrations and column burdens in air, which affect air quality and hence human exposure (WHO, 2016). Furthermore, the lifetime of these species affects their contributions to atmospheric aerosols, with a consequent influence on climate via changes to radiative transfer through scattering and cloud formation (Solomon et al., 2007). In addition, their deposition can exceed critical load thresholds causing aquatic and terrestrial acidification, and eutrophication in the case of nitrogen deposition (Howarth, 2008; Bobbink et al., 2010; Doney, 2010; Vet et al., 2014; Wright et al., 2018). Quantifying $\tau$ and deposition thus provides a crucial assessment of these regional and global impacts.

Deposition occurs through wet and dry processes. While wet deposition fluxes can be measured directly (Vet et al., 2014), there are few validated methods for dry deposition fluxes (Wesley and Hicks, 2000), and none which estimates deposition over large regions. Dry deposition fluxes ($F$) may be obtained using micrometeorological measurements for pollutants for which fast response instruments are available. However, these results are only valid for the footprints of the observation sites, typically hundreds of meters (Aubinet e al., 2012), and their extrapolation to larger regions may suffer from representativeness issues. As a result, atmospheric lifetimes $\tau$ with respect to dry deposition have not been determined through direct observations. On a regional scale, dry deposition fluxes are typically derived using an inferential

approach by multiplying network-measured or model-predicted air concentrations with dry
deposition velocities ($V_d$) (Sickles and Shadwick, 2015; Fowler et al., 2009; Meyers et al., 1991),
which are derived using resistance-based inferential dry deposition algorithms (Wu et al., 2018),
and compared with limited micrometeorological flux measurements (Wesley and Hicks, 2000;
Wu et al., 2018; Finkelstein et al., 2000; Matsuda et al., 2006; Makar et al., 2018) for validation.
When applied to a regional scale, an inferential-algorithm derived $V_d$ may have significant
uncertainties (Wesley and Hicks, 2000; Aubinet et al., 2012;Wu et al., 2018; Finkelstein et al.,
2000; Matsuda et al., 2006; Makar et al., 2018; Brook et al., 1997).  For example, inferred $V_d$ for
$SO_2$, despite being the most studied and best estimated, may be underestimated by 35% for forest
canopies (Finkelstein et al., 2000).  Underestimated $V_d$ for $SO_2$ and nitrogen oxides can
contribute to model over-prediction of regional and global $SO_2$ concentrations (Solomon et al.,
2007; Christian et al., 2015; Chin et al., 2000), or under-prediction of global oxidized nitrogen
dry deposition fluxes (Paulot et al., 2018; Dentener et al., 2006).

Here, a new approach is presented to determine $\tau$ with respect to dry deposition and $F$ for

total oxidized sulfur (**TOS**, the sulfur mass in $SO_2$ and particle-$SO_4$ (p$SO_4$)) and total reactive
oxidized nitrogen (**TON**, the nitrogen mass in NO, $NO_2$, and others designated as $NO_z$) on a
spatial scale of (3-6)x$10^3$ km$^2$, using aircraft measurements.  This approach provides a unique
methodology to determine $\tau$ and $F$ over a large region.  Coupled with analyses for chemical
reaction rates (for **TOS** compounds), the average $V_d$ for **TOS** and **TON** over the same spatial
scale were also determined.  The airborne measurements were obtained during an intensive
campaign from August to September 2013 in the Athabasca Oil Sands Region (AOSR) (Gordon
et al., 2015; Liggio et al., 2016; Li et al., 2017; Baray et al., 2018; Liggio et al., 2019) in northern
Alberta, Canada.   Direct comparisons with modelled dry deposition estimates are made to assess
their uncertainties and the spatial-temporal scales of air pollutant impacts.
**2 Methods**
**2.1 Lagrangian Flight Design**

Details of the airborne measurement program have been described elsewhere (Gordon et

al., 2015; Liggio et al., 2016; Li et al., 2017; Liggio et al., 2019; Baray et al., 2018).  Briefly, an
instrumented National Research Council of Canada's Convair-580 research aircraft was flown
over the AOSR in Alberta, Canada from August 13 to September 7, 2013.  The flights were
designed to determine emissions from mining activities in the AOSR, assess their atmospheric
transformation processes and gather data for satellite and numerical model validation.  Three
flights were flown to study transformation and deposition processes by flying a Lagrangian
pattern so that the same pollutant air mass was sampled at different time intervals downwind of
emission sources for a total of 4-5 hours and up to 107-135 km downwind of the AOSR sources.
Flights 7 (F7, Aug 19), 19 (F19, Sep 4) and 20 (F20, Sep 5) took place during the afternoon
when the boundary layer was well established.  The flights were conducted in clear sky
conditions so wet deposition processes were insignificant.  As shown in Fig. 1, the aircraft flew
tracks perpendicular to the oil sands plume at multiple altitudes between 150 to 1400 m agl and
multiple intercepts of the same plume downwind.  Vertical profiles conducted as spirals were
flown at the centre of the plume which provided information on the boundary layer height and
extent of plume mixing.  The flight tracks closest to the AOSR intercepted the main emissions
from the oil sands operations; there were no other anthropogenic sources as the aircraft flew
further downwind of the AOSR.

## 2.2 Aircraft Measurements

A comprehensive suite of detailed gas- and particle-phase measurements were made from the aircraft. Measurements pertaining to the analysis in this paper are discussed below.

**$SO_2$ and $NO_y$.** Ambient air was drawn in through a 6.35 mm (1/4") diameter PFA sampling line taken from a rear-facing inlet located on the roof towards the rear of the aircraft. The inlet was pressure-controlled to 770 mm Hg using a combination of a MKS pressure controller and a Teflon pump. Ambient air from the pressure-controlled inlet was fed to instrumentation for measuring $SO_2$ and $NO_y$. The total sample flow rate was measured at 4988 $cm^3$ $min^{-1}$ of which $SO_2$ and $NO_y$ were 429 and 1085 $cm^3$ $min^{-1}$, respectively. $SO_2$ was detected via pulsed fluorescence with a Thermo 43iTLE (Thermo Fisher Scientific, Franklin, MA, USA). $NO_y$ (also denoted as Total Oxidized Nitrogen (TON)) was measured by passing ambient air across a heated (325 ºC) molybdenum converter that reduces reactive nitrogen oxide species to NO. NO was then detected through chemiluminescence with a modified Thermo 42iTL (Thermo Fisher Scientific, Franklin, MA, USA) run in $NO_y$ mode. An inlet filter was used for $SO_2$ to exclude particles, but $NO_y$ was not filtered prior to the molybdenum converter. $NO_y$ includes NO, $NO_2$, $HNO_3$ and other oxides of nitrogen such as peroxy acetyl nitrate and organic nitrates (Dunlea et al., 2007; Williams et al., 1998). Although there was no filter on the $NO_y$ inlet to exclude particles, the inlet was not designed to sample particles (i.e. rear-facing PFA tubing). As a result, $pNO_3$ was not included as part of $NO_y$ (TON) The conversion efficiency of the heated molybdenum converter and inlet transmission was evaluated with $NO_2$ and $HNO_3$ and found to be near 100 % and >90 %, respectively. Previous studies conducted by Williams et al. (1998) showed similar molybdenum converter efficiencies including that of n-propyl nitrate near 100%. Interferences from alkenes or $NH_3$ were assumed to be negligible (Williams et al., 1998; Dunlea

et al. 2007). Species like $NO_3$ radical and $N_2O_5$ are expected to be low in concentration as they
photolyze quickly during daytime. Zeros were performed 3-5 times per flight for both the $SO_2$
and $NO_y$ instruments by passing ambient air through an in-line Koby King Jr cartridge for ~5
minutes. For the $NO_y$ measurements pre-reactor zeroes (dynamic instrument zero) were also
obtained periodically throughout each flight using either ambient air or a Koby King Jr. air
purifier. Multiple calibrations were conducted before, during and after the study using National
Institute Standards and Technology reference standards. Data were recorded at a time resolution
of 1 second and corrected for a sampling time delay of 1-3 seconds depending on the instrument.
Detection limits were determined as 2 times the standard deviation of the values acquired during
zeroes; $NO_y$ was 0.09 ppbv and $SO_2$ was 0.70 ppbv (Table S1).
**Aerosols.** Multiple aerosol instruments sub-sampled from a forward facing, shrouded, isokinetic
particle inlet (Droplet Measurement Technologies, Boulder, CO, USA). A Time-of-Flight High
Resolution Aerosol Mass Spectrometer (AMS) (Aerodyne Research Inc.) was used to measure
non-refractory submicron aerosol components including $pSO_4$, $pNO_3$, $pNH_4$, and p-organics.
Details of the AMS and its operations have been published elsewhere (DeCarlo et al., 2006).
The instrument was operated in mass spectrometry V-mode with a sampling time resolution of
10 seconds. Filtered measurements were taken 4-5 times per flight to determine background
signals. Detection limits of 0.048, 0.036, 0.235 and 0.236 ug m$^{-3}$ for $pSO_4$, $pNO_3$, $pNH_4$ and p-
organics were determined using 3 times the standard deviation of the average of filtered time
periods for all flights (Table S1). Ionization efficiency calibrations using monodisperse
ammonium nitrate were performed during the study with an uncertainty of ±9 %. Data were
corrected for a sampling time delay of 10 seconds by comparing with faster response instruments
e.g. a wing-mounted Forward Scattering Spectrometer Probe Model 300 (FSSP-300) and an in-
board Ultra High Sensitivity Absorption Spectrometer (UHSAS) (both from Droplet
Measurement Technologies).  The FSSP and UHSAS instruments measure particle diameters
that range from 300 nm-20 µm and 50 nm-1 µm, respectively.  The AMS data were processed
using AMS data analysis software (Squirrel, version 1.51H and PIKA, version 1.10H).  The
particle collection efficiency (CE) of the AMS was determined through comparisons of the total
AMS-derived mass with the mass estimated from the size distribution measurements of the
UHSAS assuming a density based on the chemical composition.  The CE for F7 and F20 was 0.5
for both flights, and for F19 it was 1.0.  The CE was applied to all AMS species for the duration
of each flight (Fig. S1).  Since the AMS measures only particle mass $< 1$ µm ($PM_1$) in diameter,
the mass of $SO_4$ formed through OH oxidation was scaled upward to account for all particle sizes
that $H_2SO_4$ vapor could potentially condense on.  The scaling factor was determined using the
surface area ratio of $PM_1/PM_{20}$ from the aircraft particle measurements, assuming that the
condensation process is approximately proportional to the surface area.  $PM_1$ measurements were
from the UHSAS and $PM_{20}$ were from the FSSP300.  As the ratio did not vary significantly in
the plumes, one single value was used between each set of screens; in F19 the ratio between
screens ranged from 0.6 to 0.8, in F20 the ratio ranged from 0.8 to 0.9, and in F7 the ratio ranged
from 0.7 to 0.9 (Liggio et al., 2016).
Measurements are discussed in terms of total oxidized sulfur (TOS, the sulfur mass in $SO_2$ from
the Thermo $SO_2$ instrument and particle-$SO_4$ ($pSO_4$) from the AMS instrument) and total
reactive oxidized nitrogen (TON, the nitrogen mass in reactive oxidized nitrogen species, from
the Thermo $NO_y$ instrument, often denoted $NO_y$).
**Volatile Organic Compounds (VOCs).**  Selected VOCs were used to estimate the OH
concentrations used for determining oxidation rates for $SO_2$.  VOCs were measured with a proton
transfer reaction time-of-flight mass spectrometer (PTR-ToF-MS, Ionicon Analytik GmbH,
Austria) as well as through discrete canister grab samples. The PTR-ToF-MS and its operation,
along with the details of the canister sampling and lab analyses during the study were described
in detail previously (Li et al., 2017). Briefly, the PTR-ToF-MS used chemical ionization with
$H_3O^+$ as the primary reagent ion. Gases with a proton affinity greater than that of water were
protonated in the drift tube. The pressure and temperature of the drift tube region were
maintained at a constant 2.15 mbar and 60 °C, respectively for an E/N of 141 Td (Townsend, 1
$Td=10^{-17}$ V $cm^2$). E/N refers to the reduced electric field parameter in the drift tube; E is the
electric field and N is the number density of the gas in the drift tube. The E/N ratio can affect the
reagent ion distribution in the drift tube and VOC fragmentation (de Gouw and Warneke; 2007).
The protonated gases were detected using a high-resolution time of flight mass spectrometer at a
time resolution of 2 seconds. Instrumental backgrounds were performed in flight using a
custom-built zero-air generating unit. The unit contained a catalytic converter heated to 350 °C
with a continuous flow of ambient air at a flow rate of one litre per minute. The data were
processed using Tofware software (Tofwerk AG). Calibrations were performed on the ground
using gas standard mixtures from Ionicon, Apel-Reimer and Scott-Marrin for 22 compounds.
The canister samples were collected in pre-cleaned and passivated 3L stainless steel canisters
that were subsequently sent to an analytical laboratory for GC-FID/MS analyses for a suite of
150 hydrocarbon compounds.
**Meteorology and aircraft state parameters.** Meteorological measurements have been
described elsewhere (Gordon et al., 2015). In brief, 3-D wind speed and temperature were
measured with a Rosemount 858 probe. Dew point was measured with an Edgetech hygrometer
and pressure was measured with a DigiQuartz sensor. Aircraft state parameters including
positions and altitudes were measured with GPS and a Honeywell HG1700 unit.  All
meteorological measurements and aircraft state parameters were measured at a 1 s time
resolution.

**2.3 Mass transfer rates in the atmosphere**

Mass transfer rates ($T$) across flight screens (Fig. 1) were determined using an extension

of the Top-down Emission Rate Retrieval Algorithm (TERRA) developed for emission rate
determination using aircraft measurements (Gordon et al., 2015).  Briefly, at each plume
interception location, the level flight tracks were stacked to create a virtual screen.  Background
subtracted pollutant concentrations and horizontal wind speeds normal to the screen were
interpolated using kriging.  The background for $SO_2$ was ~0 ppb and $pSO_4$ was 0.2-0.3 $\mu$g m$^{-3}$
which was subtracted from the $pSO_4$ measurements before mass transfer rates were calculated
(Liggio et al., 2016).  Integration of the horizontal fluxes across the plume extent on the screen
yields the transfer rate $T$ in units of t hr$^{-1}$.  Using $SO_2$ as an example,
$$T_{SO_2} = \int_{s_1}^{s_2} \int_{z_1}^{z_2} C(s,z) u_n(s,z) ds dz \qquad (1)$$
where $C(s,z)$ is the background subtracted concentration at screen coordinate s and z, which
represent the horizontal and vertical axes of the screen. The $u_n(s,z)$ is the horizontal wind speed
normal to the screen at the same coordinates.

Since the lowest flight altitude was 150 magl, it was necessary to extrapolate the data to

the surface as per the procedures described previously (Gordon et al., 2015).  Extrapolation to the
surface methods were compared and differences were included in the uncertainty estimates.  The
main sources of $SO_2$ were from elevated facility stacks associated with the desulfurization of the
raw bitumen (Zhang et al., 2018).  The stacks with the biggest $SO_2$ emissions range in height
from 76.2 to 183.0 m.  Since the main source of $SO_2$ is from the elevated facility stacks, the
uncertainty for a single screen is estimated at 4% (Gordon et al., 2015).  $NO_y$ was also
extrapolated linearly to the surface and the mass transfer rates were similarly compared to other
extrapolation methods.  $NO_y$ sources include the elevated facility stacks and surface sources such
as the heavy hauler trucks operating in the surface mines.  The uncertainty in the resulting
transfer rate $T$ for a single screen is estimated to be larger at 8 %, as a larger fraction of the $NO_y$
mass may be below the lowest measurement altitude (Gordon et al., 2015).  Sulfur and nitrogen
data were also extrapolated linearly to background values from the highest altitude flight tracks
upwards to the mixed layer height, which was determined from vertical profiles of pollutant
mixing ratios, temperature and dew point (Table 1).

Changes in the mass transfer rate $T$ (denoted $\varDelta T$) in units of t hr$^{-1}$ were then calculated as

the differences in $T$ between pairs of virtual screens.  The uncertainty in $\varDelta T$ was estimated as 8
% for **TOS** and 26 % for **TON** as supported by emission rate uncertainties determined for box
flights (Gordon et al., 2015).  The uncertainty analysis for box flights is applicable to $\varDelta T$ here, as
both account for uncertainties with an upwind and a downwind screen.  The $\varDelta T$ uncertainties
were propagated through subsequent calculations.

Knowing the change in mass transfer rate $\varDelta T$, and accounting for the net rates of

chemical loss and formation between screens for $SO_2$ and $pSO_4$, the deposition rates (and
subsequently the deposition flux in tonnes S (or N) km$^{-2}$ hr$^{-1}$, Sect. 2.4 were determined for the
sulfur compounds as follows:
$$\Delta T_{SO_2} = T_{SO_2}(t_2) - T_{SO_2}(t_1) = X_{SO2} - D_{SO2} \qquad (2)$$
$$\Delta T_{pSO_4} = T_{pSO_4}(t_2) - T_{pSO4}(t_1) = X_{pSO4} - D_{pSO4} \qquad (3)$$
$$\Delta T_{TOS} = T_{TOS}(t_2) - T_{TOS}(t_1) = -D_{TOS} \qquad (4)$$
where $X_{SO2}$ is the rate of chemical reaction loss of sulfur mass in $SO_2$, $X_{pSO4}$ is the rate of
chemical formation of sulfur mass as $pSO_4$, $D_{SO2}$ and $D_{pSO4}$ are deposition rates of sulfur mass in
$SO_2$ and $pSO_4$ respectively, and $t_1$ and $t_2$ are plume interceptions times at Screen 1 and Screen 2,
respectively. Note that the chemical loss rate of $SO_2$ is set to be equivalent to the formation rate
of $pSO_4$ ie. $X_{SO2} = X_{pSO4}$. Eq. (4) for TOS can also similarly be written as shown in Eq. (5).
$$\Delta T_{TOS} = \Delta T_{SO_2} + \Delta T_{pSO_4} = -D_{SO2} - D_{pSO4} \qquad (5)$$
Units in Eq. (2 to 5) are all in t hr$^{-1}$. Reaction with the OH radical was considered to be the most
significant chemical loss of $SO_2$ and the most significant path for the formation of $pSO_4$. $X_{SO2}$
and $X_{pSO4}$ were determined using estimated OH radical concentrations, which were estimated
using the methodology described in SI Sect. S4. Although TON encompasses a range of
different N species with expected differences in their deposition rates, it was not possible to
quantitatively separate their chemical formation/losses from their deposition rates with this
method. For total oxidized sulfur **TOS** (i.e., sulfur in $SO_2$ + $pSO_4$) and total oxidized nitrogen
**TON** (i.e., nitrogen in $NO_y$) the chemistry term is not relevant, and thus, the dry deposition rate
$D_{TOS}$ was directly determined from $\Delta T_{TOS}$ using Eq. (4), and respectively for **TON**.

**2.4 Dry deposition fluxes and dry deposition velocities**
Average dry deposition fluxes ($F$) for **TOS** and **TON** were obtained by dividing the
deposition rates $D$ in t hr$^{-1}$ with the footprint surface area of the plume between two adjacent
screens (Fig. 1 grey shaded regions), as shown in Eq. (6) for the dry deposition flux $F_{TOS}$ of **TOS**
(in t S km$^{-2}$ hr$^{-1}$):
$$F_{TOS} = \frac{D_{TOS}}{Area} \qquad (6)$$
where the surface area, *Area*, was identified as the geographic area under the plume extending to
the edges of the plume where concentrations fell to background levels (i.e. $SO_2$ to ~0 ppb; $SO_4$
~0.2 ug m$^{-3}$).  This approach was similarly used to derive deposition fluxes from an air quality
model, Global Environmental Multiscale – Modelling Air-quality and Chemistry (GEM-MaCH)
(Moran et al., 2010; also see SI Sect. S5 for details).  The geographic surface area uncertainty is
estimated at 5 %.  Dry deposition fluxes between the sources and the first screen were also
estimated using change in mass transfer rate **$\Delta T$** based on the extrapolated transfer rates back to
the source region ('extended' region).  The surface area boundaries for these 'extended' regions
were determined using latitude and longitude coordinates that were weighted by emissions.  This
was done by first using the average wind direction from Screen 1 and creating a set of parallel
back trajectories (~20) starting at different parts of Screen 1 back across the source region.  For
**TON**, the $NO_x$ emission sources along each back trajectory were weighted by their $NO_x$
emissions to obtain an emissions-weighted center location with latitude and longitude
coordinates for each back trajectory.  The line connecting these emissions-weighted center
locations formed the boundary of the extended surface area.  The extended surface area was
similarly determined for **TOS** based upon the known locations of the major $SO_2$ point sources.
The uncertainty of the 'extended' regions is estimated at 10 % based on repeated optimizations
of the geographical area.  Surface areas are visualized as grey shaded regions between screens in
Fig. 1 and tabulated in SI Table S1.

Spatially-averaged dry deposition velocities, **$V_d$**, based on the aircraft measurements were

determined over the surface area between screens using average plume concentrations across
pairs of screens at about 40 meters above the ground for **$SO_2$** and **TON** (e.g. Eq. (7) for $SO_2$ in
units of cm s$^{-1}$). Although TOS includes the S in both SO$_2$ and pSO$_4$, only SO$_2$ is used in the
calculation of $V_d$ since the deposition behaviour of gases and particles differ substantially, and
particles additionally have size-dependent deposition rates (Emerson et al., 2020). As the
dominant form of TOS is SO$_2$ (>92 %) the deposition behaviour of TOS is expected to be largely
driven by that of SO$_2$. The measured TON does not include pNO$_3$.
$$V_d = \frac{F_{SO2}}{[SO2]} \qquad\qquad (7)$$
The largest source of uncertainty in $V_d$ calculated this way was the determination of
concentration at 40 meters above the surface as the measurements were extrapolated from the
lowest aircraft altitude to the surface and interpolated concentrations were used. The
measurement-derived $V_d$ are compared with those from the air quality model GEM-MACH
which uses inferential methods.

**2.5 Monte-Carlo simulations of dry deposition velocities using multiple resistance-based**

**parameterizations**
Parameterization of dry deposition in inferential algorithms is commonly based on a
resistance approach with dry deposition velocity depending on three main resistance terms as
below:
$$V_d = \frac{1}{R_a + R_b + R_c} \qquad\qquad (8)$$

where $R_a$, $R_b$ and $R_c$ represent the aerodynamic, quasi-laminar sublayer and bulk surface
resistances respectively. Although these resistance terms are common among many regional air
quality models (Wu et al., 2018), the formulae used (and inputs in to these formulae) to calculate
the individual resistance terms differ significantly among the inferential deposition algorithms.
To assess the potential for a general underestimation of $V_d$ across different inferential deposition
algorithms, and to compare with the aircraft-derived $V_d$, five different inferential deposition
algorithms, including that used in the GEM-MACH model for calculating $V_d$ (Wu et al., 2018)
were incorporated into a Monte-Carlo simulation for $V_d$ for $SO_2$. $NO_y$ was not considered here,
as its measurement includes multiple reactive nitrogen oxide species with different individual
deposition velocities.  We note that many of the inferential algorithms are based on observations
of $SO_2$ and $O_3$ deposition made at single sites, and the extent to which a chemical is similar to
$SO_2$ or $O_3$ features into its $V_d$ calculation – the comparison thus has relevance for species aside
from $SO_2$. The five deposition algorithms considered are denoted ZHANG, NOAH-GEM,
C5DRY, WESLEY and GEM-MACH and are compared in Wu et al. (2018) (except the
algorithm in GEM-MACH).  The five algorithms all use a big-leaf approach for calculating $V_d$
i.e. $V_d$ is based on the resistance-analogy approach for calculating dry deposition velocity where
$V_d$ is the reciprocal sum of three resistance terms $R_a$, $R_b$ and $R_c$.  Although the approach is
similar, the formulations of $R_a$, $R_b$ and $R_c$ between the algorithms are substantially different
(Table 1 in Wu et al., 2018).  Results from Wu et al (2018) suggest that the differences in $R_a+R_b$
between different models would cause a difference in their $V_d$ values on the order of 10-30% for
most chemical species (including $SO_2$ and $NO_2$), although the differences can be much larger for
species with near-zero $R_c$ such as $HNO_3$.
To perform the simulations, formulae for the first four algorithms were taken from Wu et
al. (2018) and for GEM-MACH taken from Makar et al. (2018). The stomatal resistance in the
ZHANG algorithm was from Zhang et al. (2002). The GEM-MACH formula (Eq. (8.7) in the SI
of Makar et al. (2018)) for mesophyll resistance $R_{mx}$ contained a typo (missing the Leaf Area
Index (LAI)) and was corrected for as follows:
$R_{mx} = [LAI(H^*/3000 + 100\, f_0)]^{-1}$ (9)
Prescribed input values were constrained by the range of possible values consistent with the
conditions during the aircraft flights and are shown in SI Table S3 with associated references.
Calculations for the $R_a$ term were based on unstable and dry conditions as observed during the
aircraft flights. The Monte-Carlo simulation generated a distribution of possible $V_d$ values,
based on randomly generated values of the input variables to each algorithm and selected from
Gaussian distributions with a range of 3 sigma for all input parameters. All simulations were
performed with the same input values that were common between the algorithms.
**3 Results and Discussion**
**3.1 Meteorological and Emissions Conditions during the Transformation Flights**
Three aircraft flights, Flights 7 (F7), 19 (F19) and 20 (F20) were conducted in
Lagrangian patterns where the same plume emitted from oil sands activities was repeatedly
sampled for a 4-5 hour period and up to 107-135 km downwind of the AOSR. The first screen of
each flight captured the main emissions from the oil sands operations with no additional
anthropogenic sources between subsequent screens downwind. The main sources of nitrogen
oxides were from exhaust emissions from off-road vehicles used in open pit mining activities and
sulfur and nitrogen oxides from the elevated facility stack emissions associated with the
desulfurization of raw bitumen (Zhang et al., 2018).  As depicted in Fig. 1, F7 and F19 captured
a plume that contained both sulfur and nitrogen oxides.  The westerly wind direction and
orientation of the aircraft tracks on F20 resulted in the measurement of two distinct plumes; one
plume exhibited increased levels of sulfur and nitrogen oxides mainly from the facility stacks,
and the other plume contained elevated levels of nitrogen oxides, mainly from the open pit
mining activities, and no $SO_2$.

During the experiments, the dry deposition rates (**D**) (t hr$^{-1}$) were quantified under

different meteorological conditions and emissions levels of **TOS** and **TON** (***E***$_{TOS}$ and ***E***$_{TON}$) for
the three flights (see Table 1).  These differences played important roles in the observed pollutant
concentrations and resulting dry deposition fluxes for F7, F19 and F20.  Mixed layer heights
(MLH) were derived from aircraft vertical profiles that were conducted in the centre of the
plume at each downwind set of transects.  The profiles of temperature, dew point temperature,
relative humidity and pollutant mixing ratios were inspected for vertical gradients indicating a
contiguous layer connected to the surface.  The highest MLH was determined for F7 at 2500
magl whereas F19 had the lowest MLH at 1200 magl (Table 1).  In F20, the MLH was 2100
magl.  The combination of a high MLH in F7 with the highest wind speeds resulted in the lowest
pollutant concentrations of the three flights.  In F19, lower wind speeds and the lowest mixed
layer heights led to the highest pollutant levels.  F20 had emissions and meteorological
conditions that were in between F7 and F19 resulting in pollutant concentrations between those
of F7 and F19.

Emission rates of $SO_2$ and $NO_x$ (designated as ***E***$_{TOS}$ and ***E***$_{TON}$) from the main sources in

the AOSR were estimated from the aircraft measurements and varied significantly between the
three flight days. The measurement-based emission rates of ***E***$_{TOS}$ and ***E***$_{TON}$ were taken from the
mass transfer rates of $T_{SO2}$ and $T_{NOy}$ (described in Methods) by extrapolating backwards to the
source locations in the AOSR using exponential functions (Fig. 2, Sect. 3.2).  For **TOS**, the
source location was set at 57.017N, -111.466W, where the main stacks for $SO_2$ emissions are
located.  For **TON**, the source locations were determined from geographically weighted
locations.  Emission rates $E_{TOS}$ and $E_{TON}$ for each flight are shown in Table 1.
Model-based $E_{TOS}$ and $E_{TON}$ were also obtained from the 2.5 km x 2.5 km gridded
emissions fields that were specifically developed for model simulations of the large AOSR
surface mining facilities (Zhang et al., 2018) i.e. Suncor Millenium, Syncrude Mildred Lake,
Syncrude Aurora North, Shell Canada Muskeg River Mine & Muskeg River Mine Expansion,
CNRL Horizon Project and Imperial Kearl Mine.  The emissions fields have been used in GEM-
MACH (described in SI Sect. S5) to carry out a number of model simulations (Zhang et al.,
2018; Makar et al., 2018) including for the present study.  In this work, emissions were summed
from various sources including offroad, point (Continuous Emissions Monitoring (CEMS)), and
point (non-CEMS) for the surface mines to obtain total AOSR hourly emission rates for the
flight time periods of interest (Table 2).  The standard deviations reflect the emissions variations
during the simulated flight.
**3.2 Mass Transfer Rates**
The mass transfer rates $T$ (in t hr$^{-1}$) across the virtual flight screens for all three flights are
shown for **TOS** and **TON** in Fig. 1 and plotted in Fig. 2.  In F20, two distinct **TON** plumes were
observed, allowing separate $T$ calculations for **TON**.  Monotonic decreases in $T$ were observed
for both **TOS** and **TON** during transport downwind in all flights, clearly showing dry
depositional losses.  The deposition rate $D$ (Methods, Sect. 2.3) was used to estimate the
cumulative deposition of **TOS** and **TON** as a fraction of $E_{TOS}$ or $E_{TON}$ and is shown in Fig. 3 for
F7, F19 and F20 for transport distances of up to 107-135 km downwind of the sources.  Curves
were fitted to the **TOS** and **TON** dry deposition cumulative percentages from which $d_{1/e}$ and $\tau$
were determined (SI Table S1).  The transport e-folding distance ($d_{1/e}$) was determined where
63.2% of $E_{TOS}$ (or $E_{TON}$) was dry deposited, i.e., $\sum_{d=0}^{d_{1/e}} D(d) = 0.368 E_{TOS}$.  The atmospheric
lifetimes ($\tau$) were derived as $\tau = d_{1/e}/u$, where $u$ was the average wind speed across the distance
$d_{1/e}$.  These estimates were compared with predictions from the regional air quality model GEM-
MACH (Makar et al., 2018; Moran et al., 2010; SI Sect. S5) using facility emission rates (Table
2).  For **TOS** during F19, (Fig. 3b, e), the observed cumulative deposition at the maximum
distance accounted for 74±5 % vs. the modelled 21 % of $E_{TOS}$. The measurements indicate that
the cumulative deposition of TOS was due mostly to $SO_2$ dry deposition where $SO_2$ was ~100 %
of TOS closest to the oil sands sources decreasing to 94 % farthest downwind.  Although the
modelled cumulative deposition of TOS was significantly lower than the observations, the
fractional deposition of $SO_2$ was similar, decreasing from ~100 % to 95 % of TOS. Fitting a
curve to $D$ and interpolating the cumulative deposition fraction to the 63.2 % $E_{TOS}$ loss leads to a
$d_{1/e}$ of 71±1 km, versus 500 km for the model prediction.  Under the prevailing wind conditions,
the observed distance indicates a $\tau$ for **TOS** of approximately 2.2 hours, whereas the model
prediction indicated 16 hours.  Large observation-based values and model prediction differences
in lifetime were also evident for the other flights (SI Table S1).  Clearly, the model predictions
significantly underestimated deposition and vastly overestimated $d_{1/e}$ and $\tau$.  The observation-
based values for $\tau$ are also lower than average lifetimes of 1–2 days for $SO_2$ and 2–9 days for
$pSO_4$ derived from global models (Chin et al., 2000; Benkovitz et al., 2004; Berglen et al., 2004),
which include the effects of wet deposition and chemical conversion for $SO_2$, thus making their
implicit residence times with respect to dry deposition even longer.
For **TON** in F19 (Fig. 3h, l), the observed cumulative deposition accounted for 49±11 %
of $E_{TON}$ at the maximum flight distance, versus 19 % predicted by the model.  Similar model
underestimates for cumulative deposition fractions were found for F7 and F20.  Extrapolating to
the 63.2% cumulative deposition fraction, $d_{1/e}$ was estimated to be 190±7 km for F19 versus a
predicted 650 km from the model, implying a $\tau$ of approximately 5.6 hours for the measurement-
based results and 23 hours for the model prediction.  Again, analogous differences for F7 and
F20 were found (SI Table S1).  Similar to TOS, the measurement-based $d_{1/e}$ and $\tau$ values for
TON were significantly smaller than commonly accepted lifetimes of a few days for nitrogen
oxides in the boundary layer (Munger et al., 1998).
**3.3 Dry Deposition Fluxes $F$**
Using the deposition rate $D$ (in tonnes S or N hr$^{-1}$), the average dry deposition fluxes, $F$
(in tonnes S or N km$^{-2}$ hr$^{-1}$), were calculated by dividing $D$ by the plume footprint surface areas
estimated by extending to the plume edges where the concentrations fell to background levels
(Methods, Sect. 2.4).  These footprints are shown as the gray shaded geographic areas in Fig.  1,
totaling 3500, 5700 and 4200km$^2$ for F7, F19, and F20 plumes, respectively; see SI Table S1 for
**TON** plume areas).  Fig. 4a shows $F_{TOS}$ values for all three flights, exhibiting exponential
decreases with increasing distance away from the sources and showing e-folding distances for
$F_{TOS}$ of 18, 27, and 55 km for F7, F19, and F20, respectively.  More than 90% of the decreases in
$F_{TOS}$ were accounted for by $F_{SO2}$.  Similarly, $F_{TON}$ decreased exponentially with increasing
transport distances in all flights (Fig. 4c), exhibiting e-folding distances of 18 and 33 km for F7
and F19, and 55 and 189 km for the south and north **TON** plumes during F20, respectively.
These e-folding distances were similar to those for $F_{TOS}$, indicating similar rates of decreases in
$F_{TON}$ with transport distances.
The potential for other processes to contribute to the derived TOS and TON fluxes were
considered including losses from the boundary layer to the free troposphere and re-emission of
TOS or TON species from the surface back to the gas-phase.  Two different approaches, a finite
jump model and a gradient flux approach (Stull, 1988; Degrazia et al., 2015), were used to
estimate the potential upward loss across the interface between the boundary layer and the free
troposphere for sulfur and nitrogen.  In both approaches, the upward S flux was a minor loss at <
45 g km$^{-2}$ hr$^{-1}$, about 3 orders of magnitude lower than the several to many kg km$^{-2}$ hr$^{-1}$
horizontal advectional transport that were determined using TERRA.  For N, the upward flux
was estimated to be ~570 g km$^{-2}$ hr$^{-1}$, so although a larger flux than S, it is about factor of 18
lower than the TON fluxes derived from observations.
As expected from the $\tau$ and transport e-folding distance $d_{1/e}$ comparisons, the GEM-
MACH model $F_{TOS}$  were significantly lower than the measurement-based $F_{TOS}$ results (Fig. 4a),
with the model $F_{TOS}$ e-folding distances usually large: 133, 797, and 57 km for F7, F19, and F20,
respectively, or 7.4, 29.5, and 1.1 times longer than the corresponding measurement results.  Part
of the differences between model and measurement $F_{TOS}$ could be explained by differences in
actual versus model emissions, $E_{TOS}$ (Tables 1 vs 2).  To remove the influence of emissions, an
emission-normalized flux (=$F_{TOS}$/$E_{TOS}$ and $F_{TON}$/$E_{TON}$) was calculated for both measurement and
model (SI Fig. S2). Fig. 4b shows the ratios of measurement to model normalized emissions for
TOS.  The model emission-normalized fluxes $F_{TOS}$/$E_{TOS}$ were lower than the measurement-
based values by factors of 2.5-14, 1.8-3.4, and 2.0-3.0 for F7, F19, and F20, respectively,
decreasing with increased transport distances.  However, they coalesce to a factor of 2 at the
furthest distances sampled by the aircraft, indicating that the model $F_{TOS}$ estimates were biased
low by similar factors.  The decreasing trends suggest that at distances further downwind, model
fluxes may exceed measurement-based fluxes, albeit at magnitudes lower than those shown in
Fig. 4a, which is consistent with earlier study results (Makar et al., 2018).  For $F_{TON}$, the model-
predicted values were also lower than the measurement results, especially near the sources (Fig.
4c), and showed little variation with transport distances from the oil sands sources for all flights,
in strong contrast to the exponential decays observed from the aircraft.  However, the emission-
normalized fluxes ($=F_{TON}/E_{TON}$) for the model approached those from measurements within
maximum flying distances for F19 and F20, although still significantly lower for F7 (>10x) (Fig.
4d).

**3.4 Dry Deposition Velocities $V_d$**

The shorter $d_{1/e}$ and $\tau$, and larger deposition fluxes $F$ near the sources determined from
the aircraft measurements compared to predictions by the GEM-MACH model indicate that the
model dry deposition velocities $V_d$ was underestimated.  Gas-phase $V_d$ in the model is predicted
with a standard inferential "resistance" algorithm (Wesley, 1989; Jarvis, 1976), with resistance to
deposition calculated for multiple parameters including aerodynamic, quasi-laminar sublayer and
bulk surface resistances (Baldocchi, 1987).  To demonstrate the model underestimation in $V_d$,
comparisons between the measurement-based and model $V_d$ were made where an evaluation of
$V_d$ for **TOS** and **TON** was possible.  All $F_{SO2}$ were converted into $V_{d\text{-}SO2}$ by dividing $F_{SO2}$ by
interpolated $SO_2$ concentrations at 40 meters above ground, averaging 1.2±0.5, 2.4±0.4, and
3.4±0.6 cm s$^{-1}$ for F7, F19 and F20, respectively, across the plume footprints (Methods, Sect. 2.4
and SI Table S2).  The corresponding model $V_{d\text{-}SO2}$ derived in the same way as the observations
was 0.72, 0.63, and 0.58 cm s$^{-1}$, 1.7-5.4 times lower than observations (SI Sect. S5; SI Table S2).
Interestingly, the median $V_d$ for $SO_2$ of 4.1 cm s$^{-1}$ determined using eddy covariance/vertical
gradient measurements from a tower in the AOSR is higher than the mass balanced method
showing an even larger discrepancy compared to the model (SI Sect. S3; Fig. S5).  Similarly,
derived $V_{d\text{-}TON}$ averaged 2.8±0.8, 1.6±0.5, 4.7±1.4 and 2.2±0.7 cm s$^{-1}$ F7, F19, F20 south plume,
and F20 north plume, respectively (SI Table S2), 1.2-5.2 times higher than the corresponding
modelled $V_{d\text{-}TON}$ of 1.4, 1.3, 0.92, and 0.90 cm s$^{-1}$.

Using the observations, it was not possible to derive individual TON deposition rates

separate from their chemical formation/losses.  In previous modelling work, Makar et al. (2018),
use the GEM-MACH model and describe the relative contributions of different TOS and TON
species towards total S and N deposition in the AOSR.  TON was dominated by dry $NO_2$ (g)
deposition fluxes close to the sources (>70 % of total N close to the sources), and dry $HNO_3$ (g)
deposition increases with increasing distance from the sources (remaining < 30 % of total N),
and other sources of TON having minor contributions to deposition (< 10 %).  Although TON
encompasses a range of different N species with expected differences in their deposition rates,
comparisons of $V_{d\text{-}TON}$ with the model show, nevertheless, that overall large differences do exist.
**3.5 Monte-Carlo simulations of $V_d$ for $SO_2$**

To further demonstrate observation-model differences, $V_d$ distributions of $SO_2$ from five

common inferential dry deposition algorithms (Wu et al., 2018; Makar et al., 2018) were
determined for the conditions encountered during the flights using a Monte-Carlo approach as
described in Methods, Sect. 2.5).  Results for the $V_d$ simulations algorithms are shown in Fig. 5a.
Histograms for all five algorithms have peak $V_d$ values at ~1 cm s$^{-1}$ or lower.  Probability
distributions for the individual resistance terms, $R_a$, $R_b$, and $R_c$ showed that the dominant
resistance driving $V_d$ was the $R_c$ term (SI Fig. S3).  Also shown in Fig. 5a are the measurement-
derived $V_d$ for Flights 7, 19 and 20, and that from the Oski-ôtin ground site.  The observed $V_d$
values are larger than the $V_d$ values for most of the simulations, with the exception of Flight 7,
where the Zhang et al. (2002), NOAH-GEM (Wu et al., 2018) and C5DRY (Wu et al., 2018)
algorithms' distributions agree with the observations.  All algorithms are biased low relative to
the observations for the remaining flights, and the Oski-ôtin ground site.   It is noted that the
ground-site observations that were derived using a standard flux tower methodology (SI Sect.
S3) at a single site, appeared to be higher than all other $V_d$; nevertheless, these observations are
closer to the aircraft values than the algorithm estimates.  These results indicate that an
underestimation of $V_d$ relative to both aircraft and ground based measurements in the AOSR is
not unique to the GEM-MACH model or its dry deposition algorithm; similar results would
occur with the other algorithms included in the Monte-Carlo simulations, all of which are used
within other regional models.

To investigate the possible reasons behind the low model $V_d$ relative to the observations,

a series of sensitivity tests using $SO_2$ were conducted.  Differences in model $V_d$ have been shown
to be mainly due to differences in the calculated $R_c$ (Wu et al., 2018), and sensitivity tests here
indicated that $R_c$ is particularly sensitive to the cuticular resistance $R_{cut}$.  Hence, factors causing
$R_{cut}$ to change can have significant impact on model $V_d$.  In some of the algorithms, $R_{cut}$ and
other resistance terms are dependent on the effective Henry's Law constant $K_H^*$ for $SO_2$.  The
Monte-Carlo simulations for Fig. 5 assumed a surface pH= 6.68 resulting in a $K_H^*$ of $1 \times 10^5$ for
$SO_2$. Additional Monte-Carlo simulations were performed for the GEM-MACH dry deposition
algorithm by adjusting $K_H^*$ assuming different pH with small variations from a pH=6.68
significantly changing $R_c$, $R_{cut}$, and $V_d$  (SI Fig. S4). In Fig. 5b – red dashed line, with a surface
pH change from 6.68 to 8, consistent with possible alkaline surfaces in the AOSR (Makar et al.,
2018), in the GEM-MACH simulation, the $V_d$ distribution is moved to larger values) with its
peak value shifting from 0.6 to 1.4 cm s$^{-1}$.  These results show that model $V_d$ may be highly
sensitive to assumed surface pH, at least when using some inferential dry deposition algorithms
which are pH-dependent.  However, Fig. 5b shows that this pH-associated increase in $V_d$ is still
insufficient to encompass the range of measurement-derived $V_d$.  Increasing pH to 8 for the
GEM-MACH simulation reduces $R_{cut}$, hence $R_c$, to values much smaller than $R_a$ and $R_b$,
suggesting that model $V_d$ cannot further increase without reductions in both $R_a$ and $R_b$.  In other
words, $R_a$ and $R_b$ were probably overestimated in the current deposition velocity algorithms.  By
using the Zhang et al. (2002) $R_a$ and the NOAH-GEM (Wu et al., 2018) $R_b$ parameterizations in
the GEM-MACH algorithm, a further shift of the GEM-MACH $V_d$ distribution to larger values
was found, with the range encompassing most of the observations (Fig. 5b, pink dashed line).
Using the Zhang and NOAH-GEM parameterizations, rather than the GEM-MACH
parameterization, would decrease the $R_a$ and $R_b$ for the momentum, heat and moisture fluxes as
well, but still remain within the range of what is expected based on published parameterizations
(Wu et al., 2018 and references therein).

The potential for re-emission of TOS and TON species was also considered.  Fulgham et

al. (2020) report that the bidirectional fluxes of volatile organic acids are driven by an
equilibrium partitioning between surface wetness and the atmosphere.  The observations
presented here represent the net flux of all processes including the effects of deposition and any
potential re-emissions of TOS and TON compounds should this process occur.  As the results
show a net downward flux (i.e. net deposition), if any re-emission was occuring, it would be
smaller than the deposition fluxes observed here, which are themselves higher than shown by
currently available deposition algorithms.  This implies that the deposition part of the flux must
be even larger than the net observed flux and the measured net fluxes presented here should then
be considered as minimum values.  The current deposition algorithms do not include
bidirectional fluxes for inorganics, and adjustments related to pH in some situations may not be
sufficient to parameterize deposition fluxes.  A bidirectional approach may be needed that would
include not only [$H^+$], but surface heterogeneous reactions, to determine near-surface equilibrium
concentrations of co-depositing gases such as ammonia and nitric acid.

It is clear from the Monte-Carlo simulations for $SO_2$ $V_d$ comparisons, inferential dry

deposition algorithms as used in regional and global chemical transport models need to be further
validated and improved, especially over large geographic regions.  Here, the role of pH was
identified for improvement in some algorithms along with possible improvement in aerodynamic
and quasi-laminar sublayer resistance parameters.  Yet, for other algorithms and for **TON**
compounds, the model low-biases in $V_d$ remain to be investigated.

The underestimates suggest that the applications of these algorithms in regional or global

models may significantly underestimate predictions of **TOS** dry depositional loss from the
atmosphere.  Underestimates in $V_d$ are the result of a combination of uncertainties in the
parameterizations of each algorithm.  In the case of the algorithm used in GEM-MACH, by
adjusting the assumed surface pH from 6.68 to 8 (justifiable given the considerable dust
emissions in the region (Zhang et al., 2018)), the model $V_d$ moved closer to the aircraft-derived
values (Fig. 5b), reducing the model-observation gap by approximately 2/3.  In addition,
substituting the aerodynamic resistance and quasi-laminar sublayer resistance parameterizations
in the GEM-MACH algorithm with that from Zhang et al. (2002) and NOAH-GEM (Wu et al.,
2018), respectively, resulted in a further increase in the model $V_d$ distribution that encompasses
most of the observations (Fig. 5b).  Clearly, different algorithms respond differently to changes
in the parameterizations, and validation and adjustment to each algorithm needs measurement-
based results over large regions such as derived here.

## 4 Conclusions

The atmospheric transport distances and lifetimes $d_{1/e}$ and $\tau$ determined from the aircraft measurements are substantially shorter than the GEM-MACH model predictions, and the dry deposition fluxes $F$ and velocities and $V_d$ near sources are larger compared to the predictions by GEM-MACH and five inferential dry deposition velocity algorithms, respectively. There are important implications for these measurement-model discrepancies. Such discrepancies indicate that regional or global chemical transport models using these algorithms are biased low for local deposition and high for long-range transport and deposition, and **TOS** and **TON** loss from the atmosphere are significantly under-predicted, resulting in overestimated lifetimes. While the measurements took place over a relatively short time period, these results indicate that TOS and TON may be removed from the atmosphere at about twice the rate as predicted by current atmospheric deposition algorithms. This, in turn, implies a potentially significant impact on deposition over longer time scales (potentially weeks to months) and relevance towards cumulative environmental exposure metrics such as critical loads and their exceedance. A faster near-source deposition velocity for emitted reactive gases may imply less S and N mass being available for long range transport, reducing concentrations and deposition further downwind. The near-source higher deposition velocity, thus has the important implication of a reduction in more distant and longer timescale deposition for locations further from the sources. Moreover, emissions assessed through network measurements or budget analysis of atmospheric **TOS** and **TON** (Sickles and Shadwick, 2015; Paulot et al., 2018; Berglen et al., 2004) may be underestimated due to lower $V_d$ used in these estimates, and may require reassessing the effectiveness of control policies. Shorter $\tau$ for **TOS** and **TON** reduces their atmospheric spatial scale and intensity of smog episodes, potentially reducing human exposures (Moran et al., 2010).

Importantly, shorter $\tau$ for **TOS** and **TON** reduces their contribution to atmospheric aerosols;
consequently, the negative direct and indirect radiative forcing from these sulfur and nitrogen
aerosols are reduced, reducing their cooling effects on climate (Solomon et al., 2007). These
impacts suggest that more measurements to determine $\tau$ and $F$ for these pollutants across large
geographic scales and different surface types are necessary to better quantify their climate and
environmental impacts in support of policy. While in the past such determination was difficult
and/or impossible, the present study provides a viable methodology to achieve such a goal.


Table 1. Average observed meteorological conditions and facility emission rates of **TOS** ($E_{TOS}$) and **TON** ($E_{TON}$), (determined from extrapolated (to distance=0) transfer rates; Figure 1) for **TOS** and **TON** during the F7, F19 and F20 flights. SP=south plume; NP=north plume.

| Flight | Date | Time (UTC) | Mean wind speed (m/s) | Mean wind direction (°) | Mixed layer height (m agl) | $E_{TOS}$ (t/hr) | $E_{TON}$ (t/hr) |
|---|---|---|---|---|---|---|---|
| 7 | Aug 19/13 | 2007–0108 | 13.0±1.0 | 256±11.7 | 2500±100 | 3.4 | 1.2 |
| 19 | Sep 4/13 | 1854–2353 | 9.5±1.9 | 218±16 | 1200±100 | 18.5 | 3.9 |
| 20 | Sep 5/13 | 1933-2436 | 8.9±1.2 | 281±11 | 2100±100 | 5.8 | 2.2 (SP) 1.2 (NP) |

Table 2. Model average meteorological conditions and facility emission rates of **TOS** ($E_{TOS}$) and **TON** ($E_{TON}$) during the F7, F19 and F20 flights as described above. SP=south plume; NP=north plume.

| Flight | Date | Time (UTC) | mean wind speed (m/s) | mean wind direction (°) | mixed layer height (m agl) | $E_{TOS}$ (t/hr) | $E_{TON}$ (t/hr) |
|---|---|---|---|---|---|---|---|
| 7 | Aug 19/13 | 2007–0108 | 12.6±0.3 | 253±5.0 | 1670±80 | 3.8 | 2.9 |
| 19 | Sep 4/13 | 1854–2353 | 8.1±1.0 | 225±4.6 | 1450±43 | 4.3 | 2.4 |
| 20 | Sep 5/13 | 1933-2436 | 9.1±0.7 | 275±1.6 | 1590±42 | 3.7 | 1.5 (SP) 0.9 (NP) |

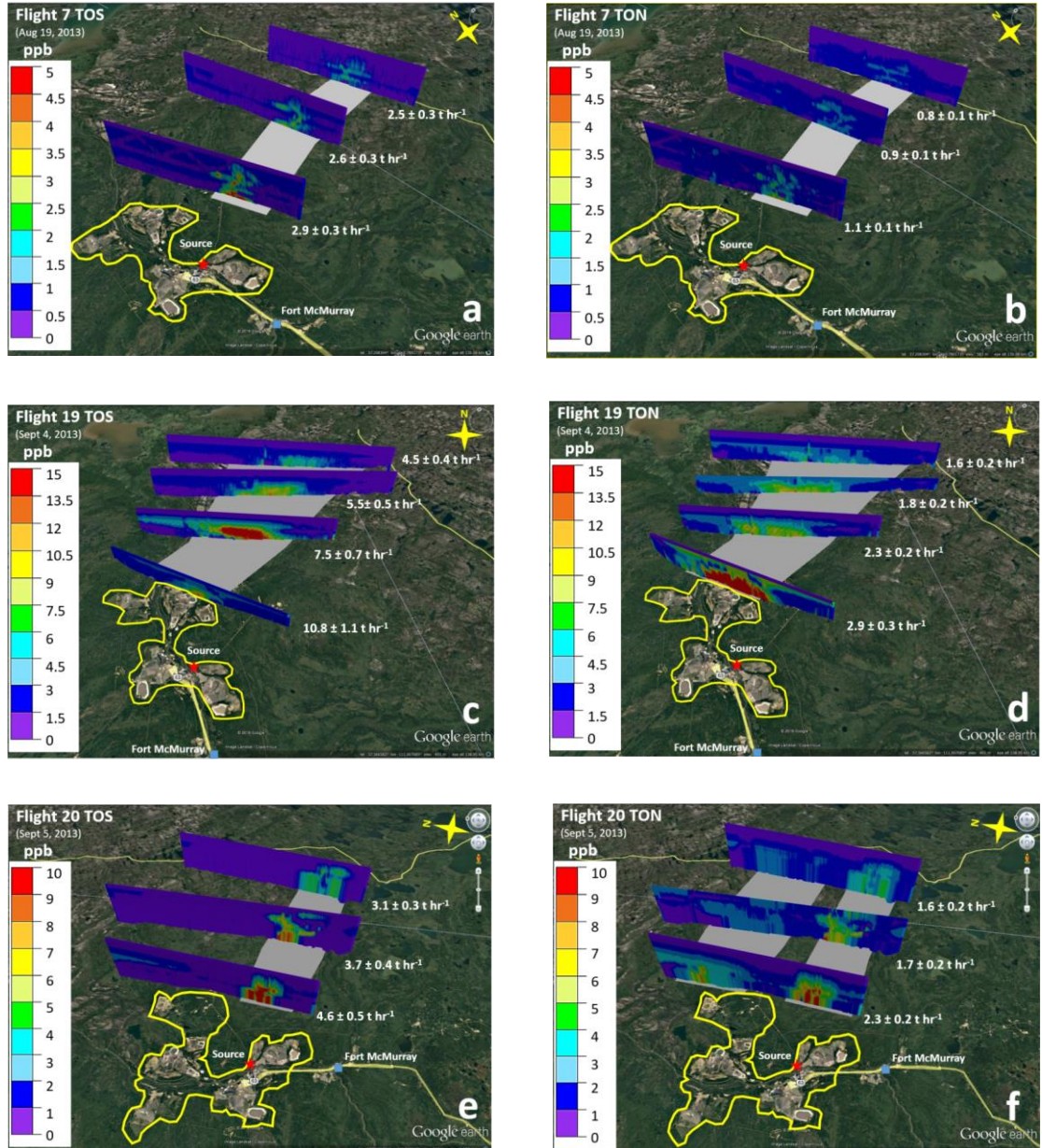

**Figure 1**. **TOS** (total oxidized sulfur) and **TON** (total oxidized nitrogen) plumes downwind of
the AOSR during three Lagrangian flights, F7, F19 and F20. The AOSR facilities are enclosed
by the yellow outline. The transfer rates $T$ in t S or N hr⁻¹ across each screen are shown. The
grey shaded surface areas are identified as the geographic footprint under the plumes. Data:
Google Image © 2018 Image Landsat / Copernicus.

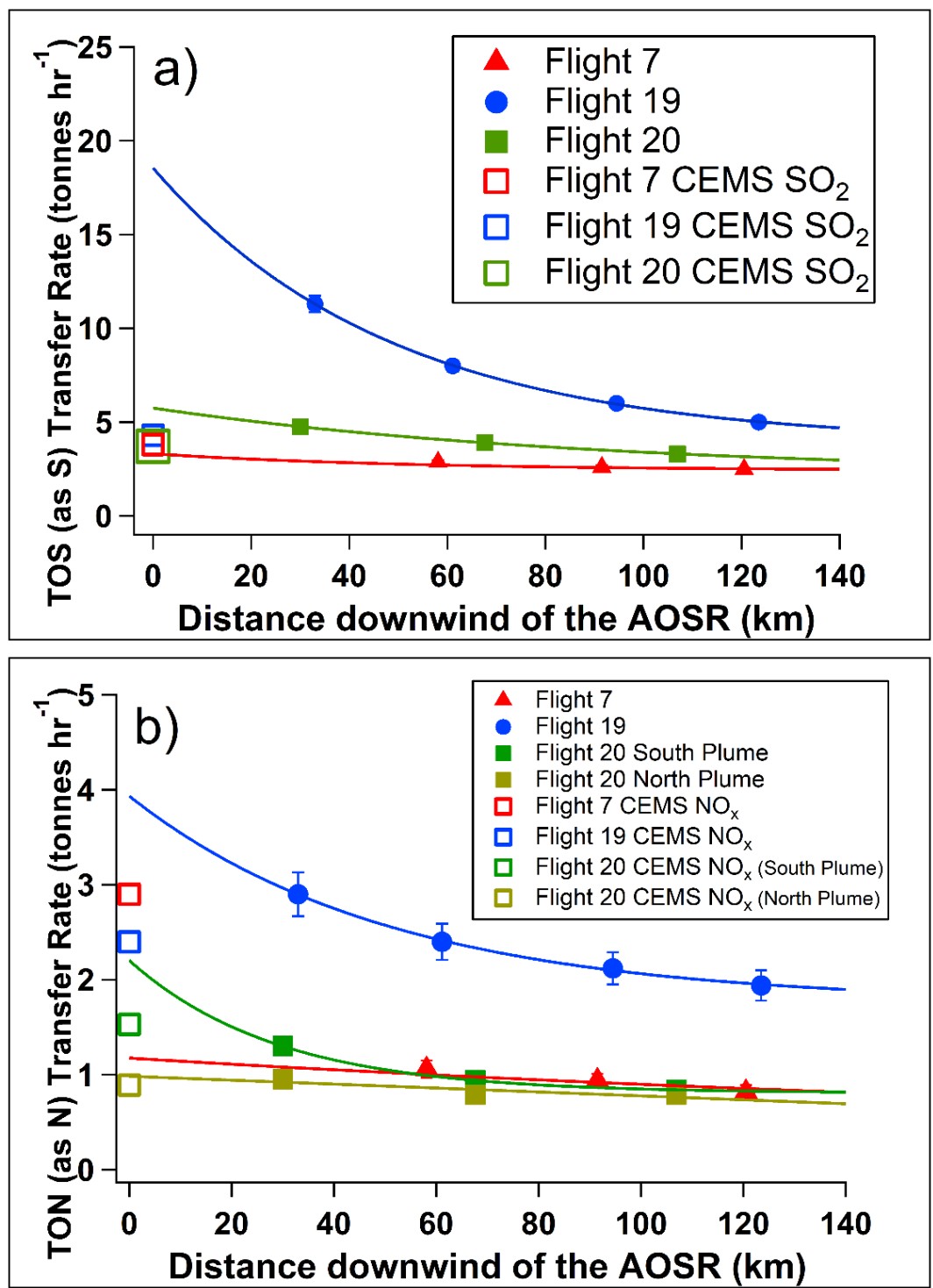


**Figure 2**. TERRA-derived transfer rates of (a) **TOS** and (b) **TON** for F7, F19 and F20. The vertical bars indicate the propagated uncertainties. The model emission rates $E_{TOS}$ and $E_{TON}$ are shown by the open symbols.

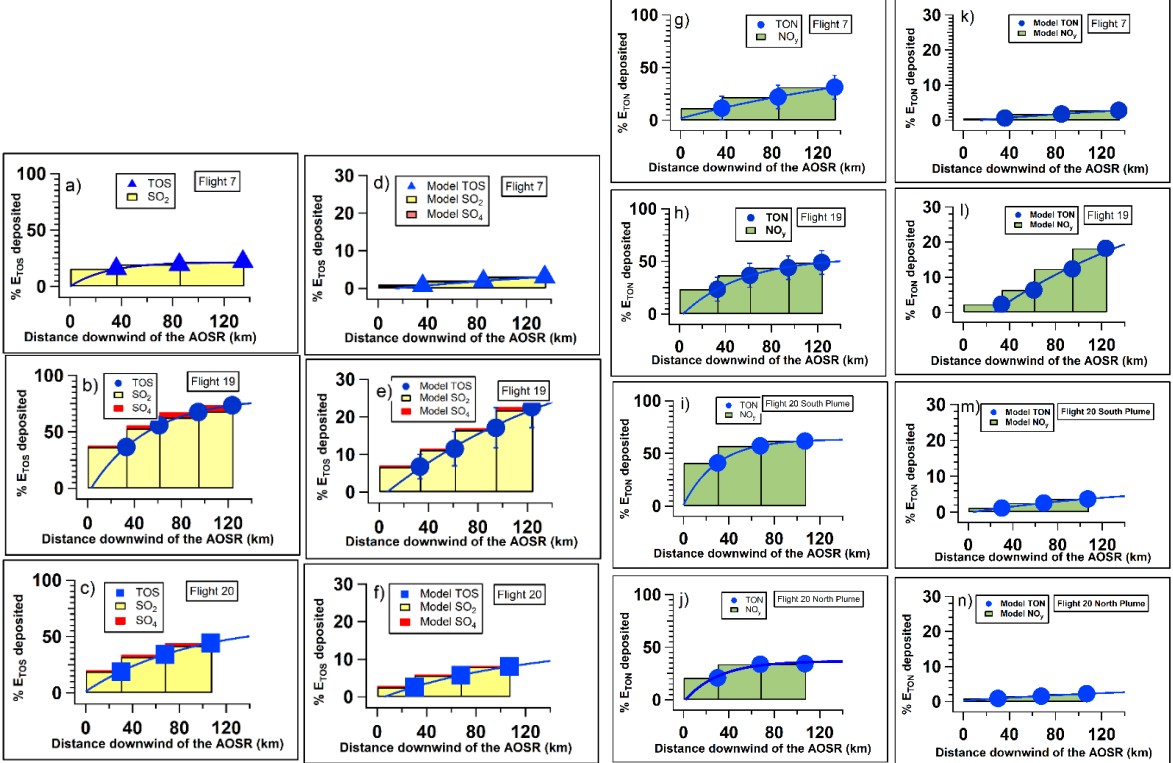


**Figure 3**. Cumulative dry deposition as a percentage of emissions $E_{TOS}$ (a to f) or $E_{TON}$ (g to n) for F7, F19 and F20 measurements with corresponding GEM-MACH model predictions. The bars show the dry deposition due to $SO_2$ and $pSO_4$. The curves were fitted to the **TOS** and **TON** dry deposition percentages from which $d_{1/e}$ and $\tau$ were determined.

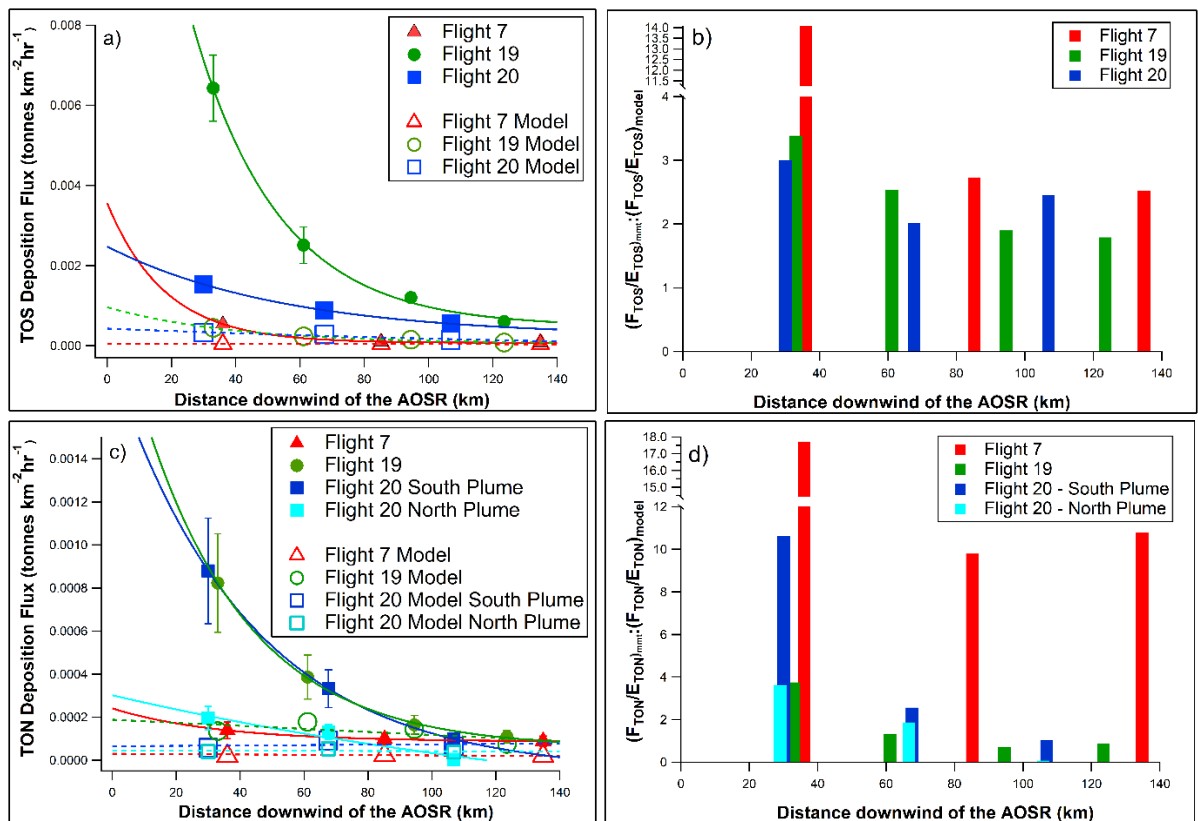

631

**Figure 4**. Dry deposition fluxes $F_{TOS}$ and $F_{TON}$ (**in t km⁻² hr⁻¹**) determined from measurements (solid symbols) and GEM-MACH model predictions (open symbols).  (a) $F_{TOS}$, (b) ratios of measurement to model normalized emissions $F_{TOS}/E_{TOS}$, (c) $F_{TON}$, and (d) ratios of measurement to model normalized emissions $F_{TON}/E_{TON}$.

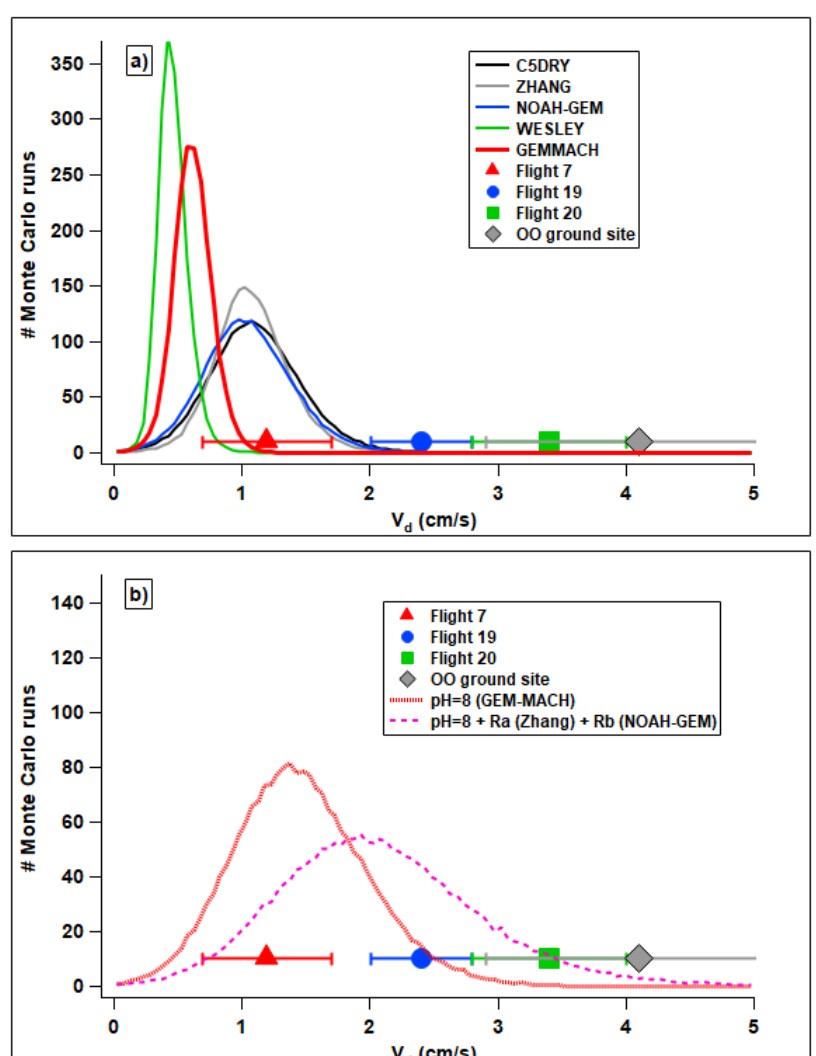

637

**Figure 5**. (a) Distributions of $V_d$ for SO$_2$ from Monte-Carlo simulations using 5 different deposition parameterizations (Wu et al., 2018; Makar et al., 2018) and (b) Monte-Carlo simulations for the GEM-MACH algorithm using a pH=8 and using a pH=8 plus replacing the GEM-MACH algorithm $R_a$ and $R_b$ formulae with that from Zhang et al. (2002) and NOAH-GEM (Wu et al., 2018), respectively. Aircraft-derived $V_d$ for F7, F19 and F20 as well as the median value for the Oski-ôtin ground site (SI Figure S5) are shown in both (a) and (b) for comparison.

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

**Code availability**

All the computer code associated with the TERRA algorithm, including for the kriging of
pollutant data, a demonstration dataset and associated documentation is freely available upon
request. The authors request that future publications which make use of the TERRA algorithm
cite this paper, Gordon et al., Liggio et al., or Li et al. as appropriate.

**Data availability**

All data used in this publication are freely available on the Canada-Alberta Oil Sands
Environmental Monitoring Information Portal: https://www.canada.ca/en/environment-climate-
change/services/oil-sands-monitoring/monitoring-air-quality-alberta-oil-sands.html

**Author contribution**

KH, SML, JL, SM, RM, RS, JO, MW all contributed to the collection of aircraft observations in the field.
KH, RM and JO made the $SO_2$, $NO_y$ and $pSO_4$ measurements and carried out subsequent QA/QC of data.
RM analyzed canister VOCs and provided OH concentration estimates. SM provided OH estimates from
MCM modelling as a comparison.  AD contributed to the development of TERRA.  JL wrote the Monte
Carlo code. PM and AA ran the model and provided model analyses. JZ provided emissions data. LZ and
RS provided deposition algorithm parameters. RS made and provided the ground site deposition velocity
measurements. KH and SML wrote the paper input from all co-authors.

**Competing Interests**
The authors declare no competing interests.
**Acknowledgements**
The authors thank the National Research Council of Canada flight crew of the Convair-580, the
Air Quality Research Division technical support staff, Julie Narayan for in-field data
management support, and Stewart Cober for the management of the study. The project was
funded by the Air Quality program of Environment and Climate Change Canada and the Oil
Sands Monitoring (OSM) program. It is independent of any position of the OSM program.