# Peer review of "New Methodology Shows Short Atmospheric Lifetimes of"

_Atmospheric Chemistry and Physics, 2020_

## Referee Comment (RC1)

**Summary/recommendations**

This is a generally interesting paper that shows promise to be of use to the community in providing a new method for tackling the poorly-constrained problem of dry deposition. However, there are a significant number of details missing in the methodology that lead me to be unable to fully evaluate the accuracy and use of the measurement techniques presented in this work. I lay out these concerns and then comment on a number of smaller issues. The authors may have internally considered and accounted for each of these issues, and if so details should be provided to strengthen the validity of their work. I recommend that this work be published in ACP after the following suggestions. Due to my concerns listed below, I am currently unable to recommend publication in ACP Letters, but would be willing to consider the revised manuscript as a candidate for ACP Letters.

The authors did not provide enough uncertainty analyses. I go into further details here:
- The authors do not discuss whether there are other possible sources of SOx/TOS. They mention that they include additional NOy sources but it's unclear whether those additional sources are being added in at appropriate spatial points in their analyses.
- Relatedly, have the authors considered bidirectional fluxes? See e.g. Fulgham et al., 2020. In that work it was noted that water films and droplets drive equilibrium partitioning, of particular importance for acids. As this study focuses on inorganic acids, there may be importance here. Can the authors discuss and perhaps provide some estimates of bidirectional flux's contribution to measured gas-phase concentrations throughout the flight path? It may be irrelevant in this system (fairly polluted; potentially little partitioning back to the gas phase) but worth doing the calculation and commenting on.
- The authors do not discuss partitioning between SO2/SO4 and within various N species. For example, "loss" of SO2 to SO4 may be misinterpreted as dry deposition losses. The methods used here to determine effects of partitioning should be explained further, along with any uncertainties that arise.
- The velocity deposition parameter Vd is hard to interpret, given that there are both gas- and particulate-phase species. Gases and particles deposit at different rates, and particles themselves have a size-dependent deposition rate (Emerson et al., 2020). The authors only use [SO2] in their calculation for Vd but apply this to TOS. What uncertainties are introduced by ignoring [pSO4], even though [SO2] is estimated in this work to account for ~90% of dry deposition? My guess is that the uncertainty is small-but it is worth addressing.
- Using an AMS for particle phase species does not provide the "total" particulate mass of a given species. For example, pSO4 could occur in coarse and fine-mode particles outside of the AMS measurement range. (The authors even state that this is a dusty region, which indicates that it is likely that an amount of nitrate and sulfate will be in the coarse mode.) The authors should discuss this and provide some uncertainty bounds on this issue.

- NOy is a notoriously tricky measurement(s) to make. I was not able to easily determine how the authors define NOy. They state on lines 136-138 "Measurements are discussed in terms of total oxidized sulfur (TOS, the sulfur mass in SO2 and particle-SO4 (pSO4)) and total reactive oxidized nitrogen (TON, the nitrogen mass in reactive oxidized nitrogen species, often denoted NOy)"--it's unclear if they are including pNO3 from the AMS in this definition. Chemiluminescence has known interference issues (e.g. Dunlea et al., 2007)-do these interferences impact the TON calculation made here, or are they lumped into TON and not important to separate out? It's also unclear how the chemiluminescence portion of the NOy set-up is being run and more details should be provided.
- How may deposition rates of different N species vary, and how may this contribute to overall uncertainty in the calculations for TON? The authors discuss how pH influences deposition, and the different N species being considered have different pKas. The relative pKa to the pH of the surface will impact a molecule's dissociation in solution (which in turn impacts how quickly more molecules will be pulled into solution).

Dunlea, E. J., Herndon, S. C., Nelson, D. D., Volkamer, R. M., San Martini, F., Sheehy, P. M., Zahniser, M. S., Shorter, J. H., Wormhoudt, J. C., Lamb, B. K., Allwine, E. J., Gaffney, J. S., Marley, N. A., Grutter, M., Marquez, C., Blanco, S., Cardenas, B., Retama, A., Ramos Villegas, C. R., Kolb, C. E., Molina, L. T., and Molina, M. J.: Evaluation of nitrogen dioxide chemiluminescence monitors in a polluted urban environment, Atmos. Chem. Phys., 7, 2691–2704, https://doi.org/10.5194/acp-7-2691-2007, 2007.

Fulgham, S. R., Millet, D. B., Alwe, H. D., Goldstein, A. H., Schobesberger, S., & Farmer, D. K. (2020). Surface wetness as an unexpected control on forest exchange of volatile organic acids. Geophysical Research Letters, 47, e2020GL088745. https://doi.org/10.1029/2020GL088745

Emerson, E. W., Hodshire, A. L., DeBolt, H. M., Bilsback, K. R., Pierce, J. R., McMeeking, G. R. and Farmer, D. K.: Revisiting particle dry deposition and its role in radiative effect estimates, Proc. Natl. Acad. Sci. U. S. A., 117(42), 26076–26082, doi:10.1073/pnas.2014761117, 2020.

**General comments**

Section 2.2 Since the work pins upon NOy and SO2, I recommend including limits of detection for each species and for relevant particulate AMS species.

Line 112-113: "Other NOy species are expected to be greater than that of HNO3." Do the authors mean greater as in concentration or greater as in conversion efficiency? Please clarify this statement. Similarly, clarify what is meant by low in "Species like NO3 radical and N2O5 are expected to be low" (line 113).

Line 132-135: The discussion about the collection efficiency of the AMS is a little vague. I suggest adding supplementary information-perhaps a figure showing time-varying CE. I assume the same CE was applied to each AMS species?

Line 156-160: I suggest including the time resolutions of the met/aircraft state parameters.

179: what elevation (height) are the stacks at?

Methods: Please provide units for each quantity/equation you introduce (especially as you are not using SI base units). It was confusing to follow along with each equation. How did you arrive at the units of dry deposition flux F, equation 5, to be t S $km^2$ $hr^{-1}$? Should this be t S $km^{-2}$ $hr^{-1}$, assuming $D_{TOS}$ to be in units of t/hr and area to be in units of $km^2$? I believe this is a typo and that you meant $km^{-2}$. As verification, eqn 6 for dry dep velocity only works if the dry deposition flux has units of t $km^{-2}$ $hr^{-1}$. Providing and checking through units for each metric should avoid any typographical of this sort. Note that this unit for F comes again on line 361.

Line 257: I suggest providing a very brief description (1-2 sentences) on what an inferential algorithm is. This will make the paper more accessible to a broader audience.

Line 266: "To assess the potential for a general underestimation of Vd across different inferential deposition algorithms..." Why do the authors assume a model underestimation in Vd a priori?

Results & discussion section: the authors heavily rely on variable abbreviations (e.g. T, TOS, TON, $E_{TOS}$, $E_{TON}$, etc) throughout this section. I suggest using words frequently to remind the reader what each variable means & to improve readability. For instance, line 361 "D" could be redefined in words.

Line 326: Why did flight F20 only show 2 distinct plumes for TON, and not TOS?

Lines 338-340: I found this statement to be a little confusing. Is the 92% dry deposition of SO2 supposed to be for the observed cumulative deposition? I suggest rewording to be more clear.

Section 3.5 This is a really interesting analysis. Did the authors run any model simulations using an adjusted Vd (say by changing pH) to see how much improvement is made in other variables studied in this work, such as TOS and $F_{TOS}$? This may equate to a significant amount more of work but would make the paper stronger. Alternatively, the authors may consider speculating/rough calculations on estimated model improvement of other variables by improved Vd.

Results in context for a longer timescale? What is total time scale of measurements (e.g. X hours after emission)

**Technical comments**
Line 76: First mention of AOSR, needs to be defined here

Line 86: 'determine emissions'... from? The oil sands specifically or the entire region? I suggest being more explicit here to help define the scope of the field campaign.

Line 147: Define E/N and Td.

**Figures and tables**
Figure 1: The resolution of the figure is rather low. I suspect this will be fixed in the final version but in the meantime, the color bars are hard to read. I recommend defining TOS and TON in the figure caption.

Figure 3: The figure caption needs more information. Explain what each grouping of subplots mean.

---

## Author Comment (AC1)

Dear Editor,

We thank the referee for their valuable comments and have replied to each question/comment below. Our responses are shown in green text and any changes made to the manuscript are italicized. The line numbers are in reference to the revised manuscript (unless stated otherwise).

Thank you for considering this manuscript for publication in ACP.

**Response to Referee #1**

The authors did not provide enough uncertainty analyses. I go into further details here:
- The authors do not discuss whether there are other possible sources of SOx/TOS. They mention that they include additional NOy sources but it's unclear whether those additional sources are being added in at appropriate spatial points in their analyses.

Thank you for your comment. In fact, we did indicate that 'there were no other anthropogenic sources downwind of the AOSR' at Line 96 (original) and provided a description of the emission sources at Line 179-183 (original). However, we agree that further clarity regarding emissions of nitrogen and sulfur oxides and their sources would be helpful. Note that Screen 1 (closest to the oil sands facilities) captured the main sources of nitrogen and sulfur oxides and there were no additional anthropogenic sources further downwind.

At lines 97-99 we included an additional sentence "*The flight tracks closest to the AOSR intercepted the main emissions from the oil sands operations; there were no other anthropogenic sources as the aircraft flew further downwind of the AOSR.*"

At Lines 216-219, we added: "*The main sources of $SO_2$ were from elevated facility stacks associated with the desulfurization of the raw bitumen (Zhang et al., 2018). The stacks with the biggest $SO_2$ emissions range in height from 76.2 to 183.0 m.*"

Also, at Line 354, an additional paragraph was added "*Three aircraft flights, Flights 7 (F7), 19 (F19) and 20 (F20) were conducted in Lagrangian patterns where the same plume emitted from oil sands activities was repeatedly sampled for a 4-5 hour period and up to 107-135 km downwind of the AOSR. The first screen captured the main emissions from the oil sands operations with no additional anthropogenic sources between subsequent screens downwind. The main sources of nitrogen oxides were from exhaust emissions from off-road vehicles used in open pit mining activities and sulfur and nitrogen oxides from the elevated facility stack emissions associated with the desulfurization of raw bitumen (Zhang et al., 2018). As depicted in Figure 1, F7 and F19 captured a plume that contained both sulfur and nitrogen oxides. The westerly wind direction and orientation of the aircraft tracks on F20 resulted in the measurement of two distinct plumes; one plume exhibited increased levels of sulfur and nitrogen oxides mainly from the facility stacks, and the other plume contained elevated levels of nitrogen oxides, mainly from the open pit mining activities, and no $SO_2$.*"

- Relatedly, have the authors considered bidirectional fluxes? See e.g. Fulgham et al., 2020. In that work it was noted that water films and droplets drive equilibrium partitioning, of particular importance for acids. As this study focuses on inorganic acids, there may be importance here. Can the authors discuss and perhaps provide some estimates of bidirectional flux's contribution to measured gas-phase concentrations throughout the flight path? It may be irrelevant in this system (fairly polluted; potentially little partitioning back to the gas phase) but worth doing the calculation and commenting on.

This is a good point and we thank the referee for bringing it up. While this is an excellent idea to consider for future work, we note that for an inorganic system such as this, there are additional complexities that would need to be considered beyond the work done by Fulgham et al. For example, we note that the organic acids examined in Fulgham et al are not assumed to have high or low concentration aqueous reactions that modify the $[H^+]$ concentration. Rather, the system is a type of binary mixture between water and the organic acid. In the type of surface reactions proposed here, the reacting surfaces are likely high concentration particles or inorganic films coating the vegetation surfaces. At these high concentrations, (molar to 10's of molar), non-unity activity coefficients need to be included as part of aqueous phase equilibria, in addition to reactions which convert $SO_2$ and dissolved $HSO_3^-$ (aq) to the sulphate ion. Multicomponent inorganic heterogeneous chemistry solvers are therefore required to derive the surface $[H^+]$ concentration used in deposition algorithms. However, the same solvers will also provide the near-surface concentrations of gases in equilibrium with the condensed phase as a step towards the prediction of a bidirectional flux. While we have started examining these approaches in current work, we also note that current bidirectional ammonia flux algorithms do not explicitly include high concentration inorganic heterogeneous chemistry on vegetated surfaces, but are largely predicated on the existence of an ammonium reservoir within the vegetation which may be filled or depleted depending on the ambient gas concentration and other factors. As such, significant theoretical and algorithm development work is required (and is underway in follow-up work), but is beyond the scope of this current paper. Regardless, it is important to note that sulphuric acid, the endpoint of $SO_2$ aqueous oxidation on leaf surfaces, is a much stronger acid than the organic acids studied by Fulgham et al. That is, partitioning of the acid itself is much more biased towards remaining in the dissociated condensed phase than for organic acids, which will tend to dissociate less with increasing molecular mass, and for which the partitioning may be more dominated by absorptive partitioning rather than dissociation. The processes governing a weak organic acid bidirectional flux are likely to differ from those governing a strong inorganic acid bidirectional flux. Re-emission of $SO_2$ itself is very unlikely given the strength of the acid and the high effective Henry's law constant of $SO_2$.

In the revised manuscript we acknowledge and explain the potential of bidirectional processes (referencing the Fulgham et al paper) by adding text at Lines 557-570: "*The potential for re-emission of TOS and TON species was also considered. Fulgham et al. (2020) report that the bidirectional fluxes of volatile organic acids are driven by an equilibrium partitioning between surface wetness and the atmosphere. The observations presented here represent the net flux of all processes including the effects of deposition and any potential re-emissions of TOS and TON compounds should this process occur. As the results show a net downward flux (i.e. net deposition), if re-emission was occurring, it would be smaller than the deposition fluxes observed here, which are themselves higher than shown by currently available deposition algorithms. This implies that the deposition part of the flux must be even larger than the net observed flux and the measured net fluxes presented here should then be considered as minimum values. The current deposition algorithms do not include bidirectional fluxes for inorganics, and adjustments related to pH in some situations may not be sufficient to parameterize deposition fluxes. A bidirectional approach may be needed that would include not only $[H^+]$, but surface heterogeneous*

*reactions, to determine near-surface equilibrium concentrations of co-depositing gases such as ammonia and nitric acid.*"

- The authors do not discuss partitioning between SO2/SO4 and within various N species. For example, "loss" of SO2 to SO4 may be misinterpreted as dry deposition losses. The methods used here to determine effects of partitioning should be explained further, along with any uncertainties that arise.

It is not clear to us, where in the manuscript the conversion of $SO_2$ to $SO_4$ as a depositional loss would be misinterpreted. In Section 2.3 (Mass transfer rates in the atmosphere), the method of determining the chemical losses/formation of $SO_2/SO_4$ are shown as per equations 2 and 3 and discussed including how OH concentrations are estimated. Regardless, we have added clarity in the text and an equation regarding losses/formation of $SO_2/SO_4$ at Lines 251-254 (which also addresses Referee #2's similar comment):

'*Note that the chemical loss rate of $SO_2$ is set to be equivalent to the formation rate of $pSO_4$ ie. $X_{SO2} =X_{pSO4}$. Equation 4 for TOS can also similarly be written as Equation 5.*

$$\Delta T_{TOS} = \Delta T_{SO_2} + \Delta T_{pSO_4} = -D_{SO2} - D_{pSO4} \qquad (5)$$

*Units in Equations 2 to 5 are all in t hr$^{-1}$.*'

Despite the conversion to SO4, its contribution to TOS is minimal. The partitioning in terms of deposition between $SO_2/SO_4$ is shown in Figure 3, included in the caption and discussed in Section 3.2 Mass Transfer Rates. However, we have provided additional text and information to be more explicit regarding the $SO_2/SO_4$ contribution to TOS deposition at Lines 415-420:

"*The measurements indicate that the cumulative deposition of TOS was due mostly to $SO_2$ dry deposition where $SO_2$ was ~100% of TOS closest to the oil sands sources decreasing to 94% farthest downwind. Although the modelled cumulative deposition of TOS was significantly lower than the observations, the fractional deposition of $SO_2$ was similar, decreasing from ~100% to 95% of TOS.*"

For the N species, although TON encompasses a range of different N species with expected differences in their deposition rates, we are not able to derive their individual deposition rates (their partitioning) separate from their chemical formation/losses from our observations. This prevents us from being able to confirm these relative contributions, or determine measurement-based estimates of the deposition velocities of the components of TON.

For increased clarity on this point, text has been added at Lines 259-261:

"*Although TON encompasses a range of different N species with expected differences in their deposition rates, it was not possible to quantitatively separate their chemical formation/losses from their deposition rates with this method.*"

For further discussion, we also add text at Lines 504-512

"*Using the observations, it was not possible to derive speciated TON deposition rates separate from their chemical formation/losses. In previous modelling work, Makar et al. (2018), use the GEM-MACH model and describe the relative contributions of different TOS and TON species towards total S and N deposition in the AOSR. TON was dominated by dry $NO_2(g)$ deposition fluxes close to the sources (>70% of total N close to the sources), and dry $HNO_3(g)$ deposition increases with increasing distance*

*from the sources (remaining always < 30% of total N), and other sources of TON having minor contributions to deposition (< 10%). Although TON encompasses a range of different N species with expected differences in their deposition rates, comparisons of V$_{d\text{-}TON}$ with the model show, nevertheless, that overall large differences do exist."*

Differences in the partitioning of TON species between the model and measurements may influence the comparison of V$_{dTON}$, but the comparison in a bulk sense is still useful as the results indeed show that such a difference, ie; factor of 2 exists. We, in fact, already had a statement at Line 575-576 that emphasizes the need for TON deposition velocities to be further investigated: '*Yet, for other algorithms and for **TON** compounds, the model low-biases in V$_d$ remain to be investigated.*'

- The velocity deposition parameter Vd is hard to interpret, given that there are both gas- and particulate-phase species. Gases and particles deposit at different rates, and particles themselves have a size-dependent deposition rate (Emerson et al., 2020). The authors only use [SO2] in their calculation for Vd but apply this to TOS. What uncertainties are introduced by ignoring [pSO4], even though [SO2] is estimated in this work to account for ~90% of dry deposition? My guess is that the uncertainty is small-but it is worth addressing.

The V$_d$ is determined only for SO$_2$ for the reasons the referee has stated above. In this study, since TOS is more than 92% SO$_2$, the V$_d$ for TOS is driven mostly by SO$_2$. Nevertheless, text has been added to more clearly articulate this at Line 295:

*"Although TOS includes the S in both SO$_2$ and pSO$_4$, only SO$_2$ is used in the calculation of **V$_d$** since the deposition behaviour of gases and particles differ substantially, and particles additionally have size-dependent deposition rates (Emerson et al., 2020). However, as the dominant form of TOS is SO$_2$ (>92%) the deposition behaviour of TOS is expected to be largely driven by that of SO$_2$. The measured TON does not include pNO$_3$."*

- Using an AMS for particle phase species does not provide the "total" particulate mass of a given species. For example, pSO4 could occur in coarse and fine-mode particles outside of the AMS measurement range. (The authors even state that this is a dusty region, which indicates that it is likely that an amount of nitrate and sulfate will be in the coarse mode.) The authors should discuss this and provide some uncertainty bounds on this issue.

Yes, indeed the AMS only measures submicron aerosols (~60nm to ~1μm) and that pSO$_4$ (and other components) can be present in both fine- and coarse-mode particles. We have already noted this issue in the paper and identified a method to account for the coarse-mode fraction (Liggio et al. 2016). This was done by using the surface area ratio of measured PM$_1$/PM$_{20}$ from aircraft particle instruments to upwards adjust the AMS pSO$_4$ concentrations to account for the 'missing' coarse mode. This explanation was originally provided in the Supplementary in Section S4, but is now moved to the main manuscript in the Methods section (Lines 156-165) for increased clarity and a reference is added. The text is as follows:

*"Since the AMS measures only particle mass < 1 μm (PM₁) in diameter, the mass of SO₄ formed through OH oxidation was scaled to account for all particle sizes that $H_2SO_4$ vapor could potentially condense on. The scaling factor was determined using the surface area ratio of $PM_1/PM_{20}$ from the aircraft particle measurements.  $PM_1$ measurements were from the UHSAS and $PM_{20}$ were from the FSSP300.  As the ratio did not vary significantly in the plumes, one single value was used between each set of screens; in F19 the ratio between screens ranged from 0.6 to 0.8, in F20 the ratio ranged from 0.8 to 0.9, and in F7 the ratio ranged from 0.7 to 0.9 (Liggio et al., 2016)."*

- NOy is a notoriously tricky measurement(s) to make. I was not able to easily determine how the authors define NOy. They state on lines 136-138 "Measurements are discussed in terms of total oxidized sulfur (TOS, the sulfur mass in SO2 and particle-SO4 (pSO4)) and total reactive oxidized nitrogen (TON, the nitrogen mass in reactive oxidized nitrogen species, often denoted NOy)"--it's unclear if they are including pNO3 from the AMS in this definition. Chemiluminescence has known interference issues (e.g. Dunlea et al., 2007)-do these interferences impact the TON calculation made here, or are they lumped into TON and not important to separate out? It's also unclear how the chemiluminescence portion of the NOy set-up is being run and more details should be provided.

Thank you for the comments.  With respect to $pNO_3$ data from the AMS, it is not included in $NO_y$ (i.e. TON). The $NO_y$ measurement did not have a filter to exclude particles, however, the inlet for the $NO_y$ measurement was not designed to sample particles, and as such, the $NO_y$ measurement would not include $pNO_3$. The text in Section 2.2 at Lines 113-118 under $SO_2$ and $NO_y$ has been modified to read:

*"An inlet filter was used for $SO_2$ to exclude particles, but $NO_y$ was not filtered prior to the molybdenum converter.  $NO_y$ includes NO, $NO_2$, $HNO_3$ and other oxides of nitrogen such as peroxy acetyl nitrate and organic nitrates (Dunlea et al., 2007; Williams et al., 1998).  Although there was no filter on the $NO_y$ inlet to exclude particles, the inlet was not designed to sample particles (i.e. rear-facing PFA tubing). As a result, $pNO_3$ was not included as part of $NO_y$ (TON)."*

With respect to the reference to the Dunlea et al. (2007) paper, the interference discussed in that paper is with the chemiluminescent method in that if one is trying to measure $NO_2$ using a molybdenum converter, the $NO_2$ concentrations may be overestimated due to interference from $NO_z$-type species (e.g. $HNO_3$, organic nitrates etc) being converted to $NO_2$.  This is due to additional $NO_z$ species that may be converted to NO across this converter.  For measurements made in urban areas this is sometimes ignored as the $NO_x$ concentrations are generally high and it is expected that $NO_x = NO_y$.  In this study, we measured $NO_y$ (i.e. TON) using the molybdenum converter with chemiluminescent detection).

Further details are added to more fully describe the $NO_y$ and $SO_2$ measurements from Lines 103-133 including sample flow rates, placement of the molybdenum converter, converter efficiencies and potential interferences (along with references) and detection limits and a table is added to summarize instrument measurement details (Table S1).

- How may deposition rates of different N species vary, and how may this contribute to overall uncertainty in the calculations for TON? The authors discuss how pH influences deposition, and the different N species being considered have different pKas. The relative pKa to the pH of the surface will impact a molecule's dissociation in solution (which in turn impacts how quickly more molecules will be pulled into solution).

Similar to the response regarding the Referee's comment on N partitioning above:

For the N species, although TON encompasses a range of different N species with expected differences in their deposition rates, it was not possible to derive their individual deposition rates (their partitioning) separate from their chemical formation/losses from our observations. This prevents us from being able to confirm these relative contributions, or determine measurement-based estimates of the deposition velocities of the components of TON. For increased clarity, text has been added at Lines 259-261:

*"Although TON encompasses a range of different N species with expected differences in their deposition rates, it was not possible to quantitatively separate their chemical formation/losses from their deposition rates with this method."*

For further discussion, we also add text at Lines 504-512:

*"Using the observations, it was not possible to derive individual TON deposition rates separate from their chemical formation/losses. In previous modelling work, Makar et al. (2018), use the GEM-MACH model and describe the relative contributions of different TOS and TON species towards total S and N deposition in the AOSR. TON was dominated by dry $NO_2(g)$ deposition fluxes close to the sources (>70% of total N close to the sources), and dry $HNO_3(g)$ deposition increases with increasing distance from the sources (remaining always < 30% of total N), and other sources of TON having minor contributions to deposition (< 10%). Although TON encompasses a range of different N species with expected differences in their deposition rates, comparisons of $V_{d\text{-}TON}$ with the model show, nevertheless, that overall large differences do exist."*

Differences in the partitioning of TON species between the model and measurements may influence the comparison of $V_{dTON}$, but the comparison in a bulk sense is still useful as the results indeed show that such a difference; ie factor of 2 exists. We do have a statement at Line 575-576 that emphasizes the need for TON deposition velocities to be further investigated: *"Yet, for other algorithms and for **TON** compounds, the model low-biases in $V_d$ remain to be investigated."*

**General comments**
Section 2.2 Since the work pins upon NOy and SO2, I recommend including limits of detection for each species and for relevant particulate AMS species.

Agreed. A table was included in the Supplementary (Table S1) to summarize the relevant measurement details including the detection limits, instrument, sampling time resolution and manufacturer. In addition, text was included in the main manuscript at Lines 141-143:

*"Detection limits of 0.048, 0.036, 0.235 and 0.236 ug $m^{-3}$ for $pSO_4$, $pNO_3$, $pNH_4$ and p-organics were determined using 3 times the standard deviation of the average of filtered time periods for all flights (Table S1)."*

and text was added at Lines 132-133:

*"Detection limits were determined as 2 times the standard deviation of the values acquired during zeroes; $NO_y$ was 0.09 ppbv and $SO_2$ was 0.70 ppbv (Table S1)."*

Line 112-113: "Other NOy species are expected to be greater than that of HNO3." Do the authors mean greater as in concentration or greater as in conversion efficiency? Please clarify this statement. Similarly, clarify what is meant by low in "Species like NO3 radical and N2O5 are expected to be low" (line 113).

Agreed, thank you! This has been reworded from Lines 121-124:

*"Previous studies conducted by Williams et al. (1998) showed similar molybdenum converter efficiencies including that of n-propyl nitrate near 100%. Interferences from alkenes or $NH_3$ were assumed to be negligible (Williams et al., 1998; Dunlea et al. 2007)."*

Line 132-135: The discussion about the collection efficiency of the AMS is a little vague. I suggest adding supplementary information-perhaps a figure showing time-varying CE. I assume the same CE was applied to each AMS species?

Yes, the same CE was applied to each AMS species based on comparisons with estimated mass from UHSAS size distributions. The CE used was not time-varying. Various methods were explored in applying the CE including that described by Middlebrook et al. (2012), however, the application of one CE was determined to be the best fit in comparing with the UHSAS.

Additional information has been added into the main manuscript and SI to more fully describe the AMS CE. In the main manuscript Lines from 148-156 have been modified to now read:

*"The FSSP and UHSAS instruments measure particle diameters that range from 200 nm – 20 μm and 50 nm - 1 μm, respectively. The AMS data were processed using AMS data analysis software (Squirrel, version 1.51H and PIKA, version 1.10H). The particle collection efficiency (CE) of the AMS was determined through comparisons of the total AMS-derived mass with the mass estimated from the size distribution measurements of the UHSAS assuming a density based on the chemical composition. The CE for F7 and F20 was 0.5 for both flights, and for F19 it was 1.0. The CE was applied to all AMS species for the duration of each flight (Figure S1)."*

In the SI, a figure and references were added (Figure S1) and associated text that reads:

*"**Figure S1. AMS total mass ($\Sigma$(p-Organics, $pSO_4$, $pNO_3$, $pNH_4$)) (gray points) compared with mass estimated from the UHSAS (black points) and the AMS CE-corrected mass (red points).** The particle collection efficiency (CE) of the AMS was investigated by comparing the total AMS-derived mass with the mass estimated from the size distribution measurements of the UHSAS. Number concentrations measured by the UHSAS over a size range of 60nm to 1μm (matching that of the AMS) were converted to volume concentrations using mid-point bin diameters and assuming spherical shapes. Volume concentrations were then converted to mass concentrations using densities weighted by the AMS components. A CE of 0.5 was determined for both F7 and F20, and for F19 it was 1.0. Detailed investigations and discussions on the CE of the AMS can be found in the literature (e.g. Middlebrook et al., 2012; Dunlea et al., 2009; Kleinman et al., 2008; Quinn et al, 2006)."*

Line 156-160: I suggest including the time resolutions of the met/aircraft state parameters.

The text was modified at Lines 195-197 to include the time resolution of met/aircraft state parameters:
*"All meteorological measurements and aircraft state parameters were measured at a 1 s time resolution."*

**179: what elevation (height) are the stacks at?**

We included this information at Lines 215-218: *"The main sources of $SO_2$ were from elevated facility stacks associated with the desulfurization of the raw bitumen (Zhang et al., 2018). The stacks with the biggest $SO_2$ emissions range in height from 76.2 to 183.0 m."*

Methods: Please provide units for each quantity/equation you introduce (especially as you are not using SI base units). It was confusing to follow along with each equation. How did you arrive at the units of dry deposition flux F, equation 5, to be t S $km^2$ $hr^{-1}$? Should this be t S $km^{-2}$ $hr^{-1}$, assuming $D_{TOS}$ to be in units of t/hr and area to be in units of $km^2$? I believe this is a typo and that you meant $km^{-2}$. As verification, eqn 6 for dry dep velocity only works if the dry deposition flux has units of t $km^{-2}$ $hr^{-1}$. Providing and checking through units for each metric should avoid any typographical of this sort. Note that this unit for F comes again on line 361.

Thank you for noting the units of equation 5 (now equation 6) which had a typo – the units should be t S $km^{-1}$ $hr^{-1}$; this has been corrected in the line just before the equation where it appeared and in the second instance in the first sentence 3.3 Dry deposition Fluxes F section. All of the equations including 2, 3, 4 and 5 already have the units described as they are each presented. We also added the units for clarity in the Figure 4 caption. To increase clarity in describing quantities/equations, we added units and text at various points throughout the manuscript as follows:

Line 208: added 'in units of t $hr^{-1}$,

Line 223: added 'transfer rate'

Line 235: modified to read: 'Changes in the mass transfer rate $T$ (denoted $\Delta T$) in units of t $hr^{-1}$,

Line 241: added 'the change in mass transfer rate'

Line 255: added 'Units in Equations 2 to 5 are all in t $hr^{-1}$.'

Line 269: added 'in t $hr^{-1}$'

Line 279: added 'change in mass transfer rate'

Line 295: added 'in units of cm $s^{-1}$.

Line 367: added '(t $hr^{-1}$)'

Line 401: added '(in t $hr^{-1}$)'

Line 441: added 'the deposition rate $D$ (in tonnes S or N hr$^{-1}$)'

Line 460: changed the format of the flux units to $< 45$ g km$^{-2}$ hr$^{-1}$

Line 488: added 'dry deposition velocities'

Figure 4: added '(in t km$^{-2}$ hr$^{-1}$)' in caption

Line 257: I suggest providing a very brief description (1-2 sentences) on what an inferential algorithm is. This will make the paper more accessible to a broader audience.

The inferential approach/algorithm description was provided early in the manuscript at (current) Line 55:

*"On a regional scale, dry deposition fluxes are typically derived using an inferential approach by multiplying network-measured or model-predicted air concentrations with dry deposition velocities ($V_d$) (Sickles and Shadwick, 2015; Fowler et al., 2009; Meyers et al., 1991), which are derived using resistance-based inferential dry deposition algorithms (Wu et al., 2018), and compared with limited micrometeorological flux measurements (Wesley and Hicks, 2000; Wu et al., 2018; Finkelstein et al., 2000; Matsuda et al., 2006; Makar et al., 2018) for validation."*

We added "*using an inferential approach*" in the above sentence for improved clarity.

Line 266: "To assess the potential for a general underestimation of Vd across different inferential deposition algorithms..." Why do the authors assume a model underestimation in Vd a priori?

We do not assume this a priori. It is based on the results of the measurement/model comparison that show the model deposition fluxes are lower than the measurements, and thus, we state that we will "*assess the potential*" for an underestimation of Vd. We don't think the text indicates that we are assuming an underestimate as we are using the word potential. No changes made.

Results & discussion section: the authors heavily rely on variable abbreviations (e.g. T, TOS, TON, $E_{TOS}$, $E_{TON}$, etc) throughout this section. I suggest using words frequently to remind the reader what each variable means & to improve readability. For instance, line 361 "D" could be redefined in words.

We take your point on abbreviations and have attempted to improve the readability as per our earlier response in dealing with quantities/equations. For the case at Line 361 (now Line 441), we have now included 'the deposition rate' to describe "D.

Line 326: Why did flight F20 only show 2 distinct plumes for TON, and not TOS?

We have added further clarity regarding the sources of TON and TOS. Added text at Lines 354-366:

*'Three aircraft flights, Flights 7 (F7), 19 (F19) and 20 (F20) were conducted in Lagrangian patterns where the same plume emitted from oil sands activities was repeatedly sampled for a 4-5 hour period and up to 107-135 km downwind of the AOSR. The first screen captured the main emissions from the oil sands operations with no additional anthropogenic sources at subsequent screens downwind. The main sources of nitrogen oxides were from exhaust emissions from off-road vehicles used in open pit mining activities and sulfur and nitrogen oxides from the elevated facility stack emissions associated with the desulfurization of raw bitumen (Zhang et al., 2018). As depicted in Figure 1, F7 and F19 captured a plume that contained both sulfur and nitrogen oxides. The westerly wind direction and orientation of the aircraft tracks on F20 resulted in the measurement of two distinct plumes; one plume exhibited increased levels of sulfur and nitrogen oxides mainly from the facility stacks, and the other plume contained elevated levels of nitrogen oxides, mainly from the open pit mining activities, and no $SO_2$.'*

Lines 338-340: I found this statement to be a little confusing. Is the 92% dry deposition of SO2 supposed to be for the observed cumulative deposition? I suggest rewording to be more clear.

The changed text in response to the previous comment on $SO_2$/$pSO_4$ partitioning should mitigate confusion on this point. At Line 415-420, the text added was:

*"The measurements indicate that the cumulative deposition of TOS was due mostly to $SO_2$ dry deposition where $SO_2$ was ~100% of TOS closest to the oil sands sources decreasing to 94% farthest downwind. Although the modelled cumulative deposition of TOS was significantly lower than the observations, the fractional deposition of $SO_2$ was similar, decreasing from ~100% to 95% of TOS."*

Section 3.5 This is a really interesting analysis. Did the authors run any model simulations using an adjusted Vd (say by changing pH) to see how much improvement is made in other variables studied in this work, such as TOS and $F_{TOS}$? This may equate to a significant amount more of work but would make the paper stronger. Alternatively, the authors may consider speculating/rough calculations on estimated model improvement of other variables by improved Vd.

Yes, we agree it is quite interesting, thank you. However, the GEM-MACH model structure did not allow for this change, so we were unable to implement these changes to the parameterization. The implementation of a surface pH estimation within the air quality model is a complex process involving the use of high concentration (non-ideal condition) inorganic heterogeneous chemistry, and is the subject of current research. However, the Monte Carlo simulation within our submitted paper uses the GEM-MACH and other model parameterizations, and hence gives a good estimate of the likely impact. There, we demonstrate the impact of pH changes on the deposition velocities, and hence the fluxes for TOS ($F_{TOS}$). With about a factor of 2 increase in the $V_d$, the $F_{TOS}$ would also change by a factor of 2. That is, the flux may be expressed as:

$$F_{TOS} = V_d \frac{\partial c}{\partial z} \cong V_d \frac{c_i}{\Delta z}$$

where $V_d$ is the deposition velocity, c is the concentration in units of mass / unit volume, z is the vertical coordinate, $\Delta z$ is the constant flux region at the bottom of the atmosphere (usually taken in air-quality

models to be the thickness of the lowest model layer), and $c_i$ is the model resolved concentration of the lowest model layer.  A doubling of $V_d$ thus results in a doubling of the deposition flux.  Our Monte Carlo simulations show that a doubling of the deposition velocity would result from a change in the surface pH from 6.68 to 8.0 and above.  The modified aerodynamic and quasi-laminar sublayer resistance formulations would increase the simulated deposition velocity by a factor of two, bringing the $F_{TOS}$ values in GEM-MACH (and other models) in line with observations. No changes made to the manuscript.

> Results in context for a longer timescale? What is total time scale of measurements (e.g. X hours after emission)

Thank you for these comments; we have added text to improve our context.  For the comment on longer timescales, we added text at lines 599-608:

"While the measurements took place over a relatively short time period, these results indicate that TOS and TON may be removed from the atmosphere at about twice the rate as predicted by current atmospheric deposition algorithms.  This, in turn, implies a potentially significant impact on deposition over longer time scales (potentially weeks to months) and relevance towards cumulative environmental exposure metrics such as critical loads and their exceedance.  A faster near-source deposition velocity for emitted reactive gases may imply less S and N mass being available for long range transport, reducing concentrations and deposition further downwind.  The near-source higher deposition velocity, thus has the important implication of a reduction in more distant and longer timescale deposition for locations further from the sources."

For the second question on measurement timescale, the measurements were made over a period of 4-5 hours.  This is indicated at Line 90 (new).  Additional text was added to improve clarity at Lines 88-91:

"*Three flights were flown to study transformation and deposition processes by flying a Lagrangian pattern so that the same pollutant air mass was sampled at different time intervals downwind of emission sources for a total of 4-5 hrs and up to 107-135 km downwind of the AOSR sources.*"

This is now additionally mentioned at Lines 354-366:

"*Three aircraft flights were conducted in Lagrangian patterns where the same plume emitted from oil sands activities was repeatedly sampled for a 4-5 hour period and up to 107-135 km downwind of the AOSR.*"

**Technical comments**
**Line 76: First mention of AOSR, needs to be defined here**

This has been corrected.  At Line 76, AOSR has been defined (Athabasca Oil Sands Region) and the next instance, Line 85, uses just the acronym.

Line 86: 'determine emissions'... from? The oil sands specifically or the entire region? I suggest being more explicit here to help define the scope of the field campaign.

This sentences at Line 86 has been modified to: *"The flights were designed to determine emissions from mining activities in the AOSR, assess their atmospheric transformation processes, and gather data for satellite and numerical model validation."*

**Line 147: Define E/N and Td.**

At Lines 178-182, text is added to explain these values as well as a reference is provided: '*The pressure and temperature of the drift tube region were maintained at a constant 2.15 mbar and 60°C, respectively for an E/N of 141 Td (Townsend, 1 Td=$10^{-17}$ V cm$^2$). E/N refers to the reduced electric field parameter in the drift tube; E is the electric field and N is the number density of the gas in the drift tube. The E/N ratio can affect the reagent ion distribution in the drift tube and VOC fragmentation (de Gouw and Warneke; 2007).*'

**Figures and tables**

Figure 1: The resolution of the figure is rather low. I suspect this will be fixed in the final version but in the meantime, the color bars are hard to read. I recommend defining TOS and TON in the figure caption.

Figure 1 has been improved. The resolution has been increased and the colour bars are much clearer. In addition, Figure 1 caption now reads: '***Figure 1***. ***TOS*** *(total oxidized sulfur) and **TON** (total oxidized nitrogen) plumes downwind of the AOSR during three Lagrangian flights, F7, F19 and F20. The AOSR facilities are enclosed by the yellow outline. The transfer rates **T** in tonnes S or N hr$^{-1}$ across each screen are shown. The grey shaded surface areas are identified as the geographic footprint under the plumes. Data: Google Image © 2018 Image Landsat / Copernicus.*'

Figure 3: The figure caption needs more information. Explain what each grouping of subplots mean.

We have added clarification text to Figure 3 and it now reads: '***Figure 3***. *Cumulative dry deposition as a percentage of emissions $E_{TOS}$ (a to f) or $E_{TON}$ (g to n) for F7, F19 and F20 measurements with corresponding GEM-MACH model predictions. The bars show the dry deposition due to $SO_2$ and $pSO_4$. The curves were fitted to the **TOS** and **TON** dry deposition percentages from which $d_{1/e}$ and $\tau$ were determined*".

---

## Author Comment (AC2)

Dear Editor,

We thank the referee for their valuable comments and have replied to each question/comment below. Our responses are shown in green text and any changes made to the manuscript are italicized. The line numbers are in reference to the revised manuscript (unless stated otherwise).

Thank you for considering this manuscript for publication in ACP.

**Response to Referee #2**

This paper presents an estimate of dry deposition fluxes using aircraft observations.

Such "regional" estimate provide a very useful constraint that can be used to improve the representation of dry deposition in global models with implications for both air quality and ecosystems. The observation-based estimate is compared with the deposition velocity calculated by the GEM-Mach model (TON and TOS) and by a suite of dry deposition algorithms (TOS only). The authors conclude that Vd is significantly underestimated and show that revisions to the representation of Ra, Rb, and Rc could reduce the model bias.

The study is interesting and fits very well within ACP.

However, I do have concerns regarding the robustness of some of some of the results (especially for NOy) and I am unable to recommend this study for publication in ACP without significant clarification.

Comments

1) line 111. What is the sensitivity to organic nitrogen?

If the sensitivity is low, how does it affect the conclusions of the study?

The $NO_y$ measurement is sensitive to organic nitrogen. Conversion is expected to be near 100% (e.g. Williams et al., 1998). This has been added to the text at Lines 121-123 with references.

2) line 191. Is this also a minor loss for NOy?

The upward flux is now also calculated for $NO_y$. Text was included at Line 462-464: '*For N, the upward flux was estimated to be ~570 g km$^{-2}$ hr$^{-1}$, so although a larger flux than S, it is about factor of 18 lower than the TON fluxes derived from observations.*'

3) eq (2). If I am not mistaken, the authors assume that X_SO2 = - X_pSO4, if so this should be made clear. I would also suggest to write equation (4) as

Delta T_TOS = Delta T_SO2 + Delta T_pSO2 = -D_SO2 - D_pSO4.

Yes, $X_{SO2} = X_{pSO4}$. This has now been explicitly indicated at Line 252-254. We have chosen to keep equation 4 as is, but for increased clarity, as per the Referee's suggestion, we have added Equation 5 at

Line 255: $\Delta T_{TOS} = \Delta T_{SO_2} + \Delta T_{pSO_4} = -D_{TOS} - D_{pSO4}$ (Note the referee likely meant Delta T_pSO4 rather than Delta T_pSO2.)

4) The authors mention that the region is very dusty. This suggests that some SO2 (and much HNO3) could react on dust. Since coarse SO4/NO3 are not measured by the AMS, such flux could be mistakenly counted as dry deposition. The authors need to clarify how this is accounted for.

This comment is addressed as per Referee #1's similar comment and repeated here.

Yes, indeed the AMS only measures submicron aerosols (~60nm to ~1µm) and that $pSO_4$ can be present in both fine- and coarse-mode particles. The authors did identify this issue and identified a method to account for the coarse-mode fraction. This was done by using a ratio of measured $PM_1/PM_{20}$ from aircraft particle instruments to adjust the AMS $pSO_4$ concentrations. This explanation was provided in the Supplementary in Section S4, but it now moved to the main manuscript in the Methods section for increased clarity of how this was accounted for.

The text is as follows: "*Since the AMS measures only particle mass < 1 µm ($PM_1$) in diameter, the mass of $SO_4$ formed through OH oxidation was scaled to account for all particle sizes that $H_2SO_4$ vapor could potentially condense on. The scaling factor was determined using the surface area ratio of $PM_1/PM_{20}$ from the aircraft particle measurements. $PM_1$ measurements were from the UHSAS and $PM_{20}$ were from the FSSP300. As the ratio did not vary significantly in the plumes, one single value was used between each set of screens; in F19 the ratio between screens ranged from 0.6 to 0.8, in F20 the ratio ranged from 0.8 to 0.9, and in F7 the ratio ranged from 0.7 to 0.9 (Liggio et al., 2016).*"

5) line 275 and line 453

It would be helpful to summarize the differences between the different dry deposition algorithms listed here (Table 1 of Wu et al. (2018), for instance).

Without such information, it is very difficult to understand the impact and validity of the changes in Ra and Rb recommended by the authors in the GEM-MACH model.

We agree a summary (including the Wu et al 2018 reference) would be useful and is now provided in Section 2.5, Lines 330-339:
"*The five deposition algorithms considered are denoted ZHANG, NOAH-GEM, C5DRY, WESLEY and GEM-MACH and are compared in Wu et al. (2018) (except the algorithm in GEM-MACH). The five algorithms all use a big-leaf approach for calculating $V_d$ i.e. $V_d$ is based on the resistance-analogy approach for calculating dry deposition velocity where $V_d$ is the reciprocal sum of three resistance terms $R_a$, $R_b$ and $R_c$. Although the approach is similar, the formulations of $R_a$, $R_b$ and $R_c$ between the algorithms are substantially different (Table 1 in Wu et al., 2018). Results from Wu et al (2018) suggest that the differences in $R_a+R_b$ between different models would cause a difference in their $V_d$ values on the order of 10-30% for most chemical species (including $SO_2$ and $NO_2$), although the differences can be much larger for species with near-zero $R_c$ such as $HNO_3$.*"

6) line 351. Deposition velocities vary a lot across the different members of the NOy family.

Differences in NO emissions between model and observations (Tables 1 and 2) could lead to biases in the ratio of NO to NO2 or the conversion rate of NOy to HNO3, which would impact the simulated Vd(NOy).

Careful evaluation of the O3 and NOy simulation are needed to support the authors' conclusions regarding Vd(NOy).

Responding similarly to a comment from Referee #1, we expect that the deposition rates of different N species do vary. Using the GEM-MACH model, Makar et al. (2018), describe the relative contributions of different TOS and TON species towards total S and N deposition in previous modelling estimates in the Athabasca oil sands region. TON was dominated by dry $NO_2$ (g) deposition flux close to the sources (>70% of total N close to the sources), and dry $HNO_3$(g) deposition with increasing distance from the sources (remaining always < 30% of total N, with other sources of TON having minor contributions (< 10%). Using the observations here, we calculate the deposition rates for TON. Although TON encompasses a range of different N species with expected differences in their deposition rates, it was not possible to derive their individual deposition rates separate from their chemical formation/losses from our observations. This prevents us from being able to confirm these relative contributions, or determine measurement-based estimates of the deposition velocities of the components of TON.

For increased clarity, text has been added at Lines 260-262: "*Although TON encompasses a range of different N species with expected differences in their deposition rates, it was not possible to quantitatively separate their chemical formation/losses from their deposition rates with this method.*"

For further discussion, we also add text at Lines 505-513 '*Using the observations, it was not possible to derive individual TON deposition rates separate from their chemical formation/losses. In previous modelling work, Makar et al. (2018), use the GEM-MACH model and describe the relative contributions of different TOS and TON species towards total S and N deposition in the AOSR. TON was dominated by dry $NO_2$ (g) deposition fluxes close to the sources (>70% of total N close to the sources), and dry $HNO_3$ (g) deposition increases with increasing distance from the sources (remaining always < 30% of total N), and other sources of TON having minor contributions to deposition (< 10%). Although TON encompasses a range of different N species with expected differences in their deposition rates, comparisons of $V_{d\text{-}TON}$ with the model show, nevertheless, that overall large differences do exist.*'

With regard to the comment on differences between the measurements and model amongst the TON ($NO_y$) species (e.g conversion to $HNO_3$; we think the reviewer intended to say '….the conversion rate of $NO_2$ to $HNO_3$') - such differences do indeed limit the comparisons of $V_{d\text{-}TON}$ to a bulk comparison. Differences in the partitioning of TON species between the model and measurements may influence the comparison of $V_{dTON}$, but the comparison in a bulk sense is still useful as the results indeed show that such a difference ie factor of 2 exists. We do have a statement at Line 576-577 that emphasizes the need for TON deposition velocities to be further investigated: '*Yet, for other algorithms and for **TON** compounds, the model low-biases in $V_d$ remain to be investigated.*'

7) I assume that changing Ra (and Rb?) will not only impact the removal of chemical

species but also the heat, moisture, and momentum fluxes in GEM-Mach. Could the authors discuss the magnitude of these changes?

Yes, this is true. Ra parameterizes turbulent transport in the boundary layer, which occurs through turbulent eddies and behaves similarly (but not identically) for heat, moisture and momentum. It becomes more complicated for Rb since molecular and thermal diffusion processes take over from turbulent processes. Using the Zhang and Wu parameterizations rather than those currently implemented in GEM-MACH would decrease the Ra and Rb for the momentum, heat and moisture fluxes as well, but still remain within the range of what is expected based on published parameterizations. A similar shift of probability distribution functions in terms of relative magnitude as shown in Fig. 5b) would be expected for these fluxes as well.  No changes made in the manuscript.

Minor comments:

1) line 20

"Dry deposition fluxes decreased exponentially with distance" This statement is unclear. Distance from where?

Agreed!  We have clarified by modifying the statement at Line 21 as such: '*Dry deposition fluxes decreased exponentially with distance from the Athabasca oil sands sources, located in northern Alberta, resulting in lifetimes of 2.2-26 hours*.'

2) line 250

GEM-MACH has not been introduced yet. Delete or define earlier.

GEM-MACH was actually introduced, defined (and referenced) earlier at Line 227 (now Line 277). However, for additional clarity, we added text at Line 308 so the sentence that the reviewer commented on now reads: '*The measurement-derived $V_d$ are compared with those from the air quality model GEM-MACH which uses inferential methods*.'

3) line 294

Introduce notation F7 as flight 7 (F7)

Although the flight notation was already defined earlier (in Section 2.1 Lagrangian Flight Design (current Line 91), for clarity, the notation is now repeated at Section 3.1, first line (Line 355; was Line 291).

4) fig. 1  Flight 20 shows two plumes for TON but 1 plume for TOS. Could the authors comment on this difference?

This comment is addressed as per Referee #1's similar comment and repeated here.

We have added further clarity regarding the sources of TON and TOS. Added text at Lines 355-367

*'Three aircraft flights, Flights 7 (F7), 19 (F19) and 20 (F20) were conducted in Lagrangian patterns where the same plume emitted from oil sands activities was repeatedly sampled for a 4-5 hour period and up to 107-135 km downwind of the AOSR. The first screen captured the main emissions from the oil sands operations with no additional anthropogenic sources at subsequent screens downwind. The main sources of nitrogen oxides were from exhaust emissions from off-road vehicles used in open pit mining activities and sulfur and nitrogen oxides from the elevated facility stack emissions associated with the desulfurization of raw bitumen (Zhang et al., 2018). As depicted in Figure 1, F7 and F19 captured a plume that contained both sulfur and nitrogen oxides. The westerly wind direction and orientation of the aircraft tracks on F20 resulted in the measurement of two distinct plumes; one plume exhibited increased levels of sulfur and nitrogen oxides mainly from the facility stacks, and the other plume contained elevated levels of nitrogen oxides, mainly from the open pit mining activities, and no $SO_2$.'*